



# Transferability of climate simulation uncertainty to hydrological impacts

**Hui-Min Wang**[1], **Jie Chen**[1], **Alex J. Cannon**[2], **Chong-Yu Xu**[1,3], and **Hua Chen**[1]

[1]State Key Laboratory of Water Resources and Hydropower Engineering Science, Wuhan University, Wuhan, 430072, China
[2]Climate Research Division, Environment and Climate Change Canada, Victoria BC, Canada
[3]Department of Geosciences, University of Oslo, Oslo, Norway

**Correspondence:** Jie Chen (jiechen@whu.edu.cn)

**Abstract.** `TS1` `TS2` Considering rapid increases in the number of climate model simulations being produced by modelling centres, it is often infeasible to use all of them in climate change impact studies. In order to thoughtfully select subsets of climate simulations from a large ensemble, several envelope-based methods have been proposed. The subsets are expected to cover a similar uncertainty envelope to the full ensemble in terms of climate variables. However, it is not a given that the uncertainty in hydrological impacts will be similarly well represented. Therefore, this study investigates the transferability of climate uncertainty related to the choice of climate simulations to hydrological impacts. Two envelope-based selection methods, $K$ means clustering and the Katsavounidis–Kuo–Zhang (KKZ) method, are used to select subsets from an ensemble of 50 climate simulations over two watersheds with very different climates using 31 precipitation and temperature variables. Transferability is evaluated by comparing uncertainty coverage between climate variables and 17 hydrological variables simulated by a hydrological model. The importance of choosing climate variables properly when selecting subsets is investigated by including and excluding temperature variables. Results show that KKZ performs better than $K$ means at selecting subsets of climate simulations for hydrological impacts, and the uncertainty coverage of climate variables is similar to that of hydrological variables. The subset of the first 10 simulations covers over 85 % of total uncertainty. As expected, temperature variables are important for the snow-related watershed, but less important for the rainfall-driven watershed. Overall, envelope-based selection of around 10 climate simulations, based on climate variables that characterize the physical processes controlling the hydrology of the watershed, is recommended for hydrological impact studies.

## 1 Introduction

In studies of climate change impacts on hydrology, multi-model ensembles (MMEs) formed by multiple global climate models (GCMs) and multiple emission scenarios have been widely used to drive hydrological models (Minville et al., 2008; Vaze and Teng, 2011; Mehran et al., 2014; Chen et al., 2011b). There are two strengths of using MMEs: (1) the MME mean typically performs better than any individual model in representing the mean of historical climate observations (Gleckler et al., 2008; Pierce et al., 2009; Pincus et al., 2008; Mehran et al., 2014); and (2) the spread of a MME can be used to estimate climate change impact uncertainties, for example those related to GCM structure, future greenhouse gas concentrations, and internal climate variability (Mendlik and Gobiet, 2016; Knutti et al., 2010; Chen et al., 2011b; Tebaldi and Knutti, 2007). While climate projection uncertainty and spread or coverage of a MME are not equivalent, the latter does provide an imperfect estimate of uncertainty and, for sake of simplicity, we use the terms interchangeably in the remainder of this study.

The number of GCM simulations available for impact studies is increasing rapidly. For instance, the Coupled Model Intercomparison Project Phase 3 (CMIP3) contains outputs from 25 different GCMs, whereas CMIP5 contains

outputs from 61 GCMs (https://pcmdi.llnl.gov), TS3 with each GCM contributing one or more simulation runs (Taylor et al., 2012). Although it is usually advised that as many climate simulations as possible be used in impact studies, the extraction, storage, and computational costs associated with a large MME may be prohibitive. In practice, it is not uncommon for impact studies to instead rely on a small subset of climate simulations, the members of which are often selected manually, relying on expert judgement.

Several studies have considered more objective means of selecting subsets of climate simulations for impact studies based on different criteria. Generally, there are two main types of selection approaches. The past-performance approach selects a subset by weighting or selecting simulations according to their ability to represent the observed near-past climate (Gleckler et al., 2008; Perkins et al., 2007; Pincus et al., 2008). Climate model performance is generally evaluated based on agreement with observed climate conditions, which is often defined by a suite of climate metrics. For example, Perkins et al. (2007) ranked climate models based on probability density functions of observed temperature and precipitation. Similarly, Gleckler et al. (2008) evaluated the performances of 22 GCMs according to relative errors of some climatological fields, but stressed that a wider range of metrics might give more robust results. In general, the assumption that models with good performance over the near-past provide more realistic climate change signals is questionable (Knutti et al., 2010; Reifen and Toumi, 2009), although recent work on emergent constraints suggests that it may be possible to remove models that fail to represent certain key physical processes that dictate the evolution of long-term climate projections (Klein and Hall, 2015). In practice, however, the metrics commonly used to evaluate model performance are often manually defined based on the fields of interest, which leads to substantial subjectivity within the weighting process.

Another means of selecting climate simulations is the envelope-based approach, which tries to select a representative subset of climate simulations that covers as much of the full ensemble's range of future climate change signals as possible (Warszawski et al., 2014; Cannon, 2015; Logan et al., 2011). For instance, Cannon (2015) used two automated multivariate statistical algorithms, $K$ means clustering and the Katsavounidis–Kuo–Zhang (KKZ) method (Katsavounidis et al., 1994), to select subsets of CMIP5 GCMs that bracket the overall range of changes in a suite of 27 climate extreme indices. The goal of the envelope-based approach coincides with the motivation behind the usage of a MME, namely to account for different sources of projection uncertainty, including structural uncertainty (Wilcke and Bärring, 2016; Tebaldi and Knutti, 2007).

Some studies have proposed selection methods that combine both near-past performance and climate change envelope coverage criteria (McSweeney et al., 2012; Lutz et al., 2016a; Giorgi and Mearns, 2002). For example, Lutz et al. (2016a) took both model historical skill and the range of projected changes in means and extremes into consideration through a three-step sequential selection procedure. Since many examples of such selection methods place emphasis on ranking or weighting climate model performance, they may inherit the potential flaws of the past-performance approach.

Regardless of the underlying approach, most selection methods are only conducted on climate variables that can be calculated directly from the MME simulation outputs. Even though subsets of simulations that account for most of the ensemble spread in climate variables can be identified, it is not guaranteed that the same level of spread coverage extends to hydrological impact variables due to the complexity and non-linearity of hydrological processes. For example, small perturbations in the frequency or intensity of temperature and precipitation regimes may have noticeable impacts on streamflow patterns and flood magnitudes (Muzik, 2001; Whitfield and Cannon, 2000). Consequently, whether the adequate coverage of climate simulation uncertainty is transferable to hydrological impacts should be evaluated before applying envelope-based selection methods in hydrological impact studies.

Chen et al. (2016) investigated the transferability of optimally selected climate simulations in the uncertainty quantification of hydrological impacts over a Canadian watershed. They concluded that the transferability of climate simulation uncertainty is limited to hydrological impacts. However, the selection methods used in their study were applied to just two climate variables, mean annual temperature and mean annual precipitation, which is a common strategy employed by practitioners who employ envelope-based approaches (Immerzeel et al., 2013; Warszawski et al., 2014). Hydrological responses are driven both by annual climate conditions and intra-annual climate processes, which may not be described by a small number of climate variables. The transferability of climate uncertainty may be diminished due to insufficient climate variables.

Accordingly, the main objective of this study is to investigate the transferability of climate simulation uncertainty to the assessment of hydrological impacts by using a large pool of climate variables, including seasonal means, annual means, and climate extremes. The case study is conducted over two watersheds with very different climate conditions, one of which is seasonally snow-covered and the other driven by summer monsoon rainfall with little winter snowfall. Two envelope-based approaches ($K$ means clustering and the KKZ method) are used to select subsets of climate simulations based on different sets of climate variables. Transferability is evaluated by comparing the uncertainty coverage between the climate variables and 17 hydrological variables simulated by a hydrological model.

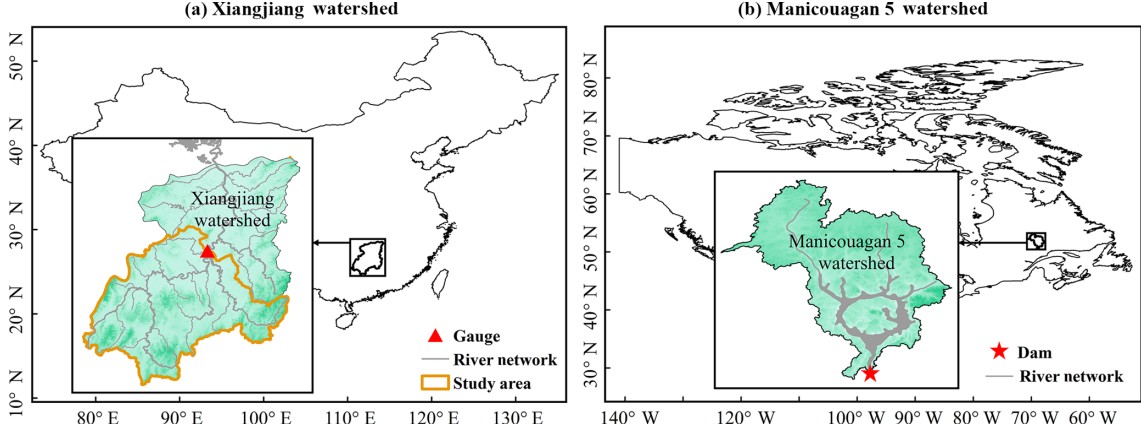

**Figure 1.** Location maps of the **(a)** Xiangjiang and **(b)** Manicouagan 5 watersheds. (The study area in the Xiangjiang watershed is one of its sub-basins, as the orange boundary shows.)

## 2 Study area and data

### 2.1 Study area

This study was conducted over two watersheds (the Xiangjiang and Manicouagan 5 watersheds) with different climate and hydrological characteristics (Fig. 1). The Xiangjiang watershed is a monsoon-climate and rainfall-dominated watershed located in southern central China, whereas Manicouagan 5 is a temperate-climate and seasonally snow-covered watershed located in central Quebec, Canada.

### 2.1.1 Xiangjiang watershed

The Xiangjiang watershed is one of the largest sub-basins of the Yangtze River watershed (Fig. 1a). The Xiangjiang River originates from the Haiyang Mountains in the Guangxi Autonomous Region and flows north to Dongting Lake in Hunan Province, which connects to the Yangtze River. The Xiangjiang River consists of several tributaries with a surface area of approximately 94 660 km$^2$, but only the watershed with an area of 52 150 km$^2$ above the Hengyang gauging station was used in this study. The watershed has a hilly topography ranging from a maximum elevation of 2042 m a.s.l. to a minimum elevation of 58 m a.s.l. at the Hengyang station. The Xiangjiang watershed is heavily influenced by a subtropical monsoon climate with hot and humid summers and mild and dry winters. The average annual precipitation over the catchment is about 1570 mm almost entirely in the form of rainfall. Around 61 % precipitation occurs from April to August, resulting in high flows during this period. The average daily maximum and minimum temperatures are around 22 and 15 °C, respectively. The average daily discharge at the Hengyang station is around 1400 m$^3$ s$^{-1}$. The peak discharge of the averaged daily hydrograph is about 4420 m$^3$ s$^{-1}$, mainly resulting from high-intensity rainfall.

### 2.1.2 Manicouagan 5 watershed

The Manicouagan 5 watershed, the largest sub-basin of the Manicouagan River watershed, is located in the centre of the province of Quebec, Canada (Fig. 1b). The Manicouagan 5 River discharges into the Manicouagan reservoir, an annular reservoir within the remnant of an ancient eroded impact crater, and ends at the Daniel Johnson Dam, which is the largest buttressed multiple arc dam in the world. The drainage area of the Manicouagan 5 River is about 24 610 km$^2$, which is mostly covered by forest and has a moderately hilly topography ranging from a maximum elevation of 952 m to a minimum elevation of 350 m a.s.l. (Chen et al., 2016). The Manicouagan 5 watershed has a continental subarctic climate dominated by long and cold winters. The annual precipitation is fairly evenly distributed within the year and averages about 912 mm, around 45 % of which is snowfall. The average daily maximum and minimum temperatures are around 2.4 and −7.8 °C, respectively. The average discharge of the Manicouagan 5 River is about 530 m$^3$ s$^{-1}$. The peak discharge of the averaged daily hydrograph is around 2200 m$^3$ s$^{-1}$, mainly resulting from snowmelt.

### 2.2 Data

Both observed and simulated daily meteorological (maximum and minimum temperatures and precipitation) data over both watersheds were used in this study. All the climate data from multiple stations or grids were averaged over the watersheds.

### 2.2.1 Climate simulations

Climate model simulation data used in this study were extracted from the CMIP5 archive (Taylor et al., 2012) for both the historical reference (1975–2004) and future (2070–2099) projection periods. Twenty-six GCMs from 15 institutions were employed to represent climate modelling uncer-

**Table 1.** Basic information about the CMIP5 models.

| Institution | Model name | Resolution (long. × lat.) |
|---|---|---|
| Commonwealth Scientific and Industrial Research Organization (CSIRO) and Bureau of Meteorology (BOM), Australia | ACCESS1.0 ACCESS1.3 | 1.875 × 1.25 1.875 × 1.25 |
| Beijing Climate Center, China Meteorological Administration | BCC-CSM1.1 BCC-CSM1.1(m) | 2.8 × 2.8 1.125 × 1.125 |
| College of Global Change and Earth System Science, Beijing Normal University | BNU-ESM | 2.8 × 2.8 |
| Canadian Centre for Climate Modelling and Analysis | CanESM2 | 2.8 × 2.8 |
| Centro Euro-Mediterraneo per I Cambiamenti Climatici | CMCC-CMS CMCC-CM CMCC-CESM | 1.875 × 1.875 0.75 × 0.75 3.75 × 3.7 |
| Centre National de Recherches Météorologiques/Centre Européen de Recherche et Formation Avancée en Calcul Scientifique | CNRM-CM5 | 1.4 × 1.4 |
| Commonwealth Scientific and Industrial Research Organization in collaboration with Queensland Climate Change Centre of Excellence | CSIRO-Mk3.6.0 | 1.8 × 1.8 |
| LASG, Institute of Atmospheric Physics, Chinese Academy of Sciences; and CESS, Tsinghua University | FGOALS-g2 | 1.875 × 1.25 |
| NOAA Geophysical Fluid Dynamics Laboratory | GFDL-CM3 GFDL-ESM2G GFDL-ESM2M | 2.5 × 2.0 2.5 × 2.0 2.5 × 2.0 |
| Institute for Numerical Mathematics | INM-CM4 | 2.0 × 1.5 |
| Institut Pierre-Simon Laplace | IPSL-CM5A-LR IPSL-CM5A-MR IPSL-CM5B-LR | 3.75 × 1.9 2.5 × 1.25 3.75 × 1.9 |
| Japan Agency for Marine-Earth Science and Technology, Atmosphere and Ocean Research Institute (The University of Tokyo), and National Institute for Environmental Studies | MIROC-ESM-CHEM MIROC-ESM | 2.8 × 2.8 2.8 × 2.8 |
| Atmosphere and Ocean Research Institute (The University of Tokyo), National Institute for Environmental Studies, and Japan Agency for Marine-Earth Science and Technology | MIROC5 | 1.4 × 1.4 |
| Max Planck Institute for Meteorology (MPI-M) | MPI-ESM-LR MPI-ESM-MR | 1.875 × 1.875 1.875 × 1.875 |
| Meteorological Research Institute | MRI-ESM1 MRI-CGCM3 | 1.125 × 1.125 1.1 × 1.1 |

tainty (Table 1). Two Representative Concentration Pathways (RCP4.5 and RCP 8.5) were used for each GCM to represent forcing scenario uncertainty, with the exception of CMCC-CESM, which only used RCP8.5, and MRI-ESM1, which only used RCP4.5. Only the first run of each GCM was used. On the whole, an ensemble of 50 climate simulations was used in this study.

### 2.2.2 Observations

Observed daily meteorological data used to downscale the GCM outputs and calibrate the hydrological model cover the 1975–2004 period for both watersheds. Meteorological data for the Manicouagan 5 watershed were obtained from the 10 km gridded dataset of Hutchinson et al. (2009), which was created by fitting spatially continuous functions of longitude, latitude, and elevation to daily station data using a trivariate thin plate smoothing spline interpolation algorithm. Discharge data at the outlet of the Manicouagan 5 River were based on mass balance calculations at the Daniel Johnson Dam. Meteorological and discharge data for the Xiangjiang watershed were observed at 97 rain gauges, 8 temperature gauges, and 1 streamflow gauge in the catchment above the Hengyang station, which are the same as those used in Zeng et al. (2016) and Xu et al. (2013).

Please note the remarks at the end of the manuscript.

**Table 2.** Definitions of 31 climate variables. The final column indicates whether the change in a given variable is expressed in the form of relative difference (CT: change type).

| Category | Index | Description | CT |
|---|---|---|---|
| ETCCDI extreme indices | TXx | Annual maximum value of daily maximum temperature | |
| | TXn | Annual minimum value of daily maximum temperature | |
| | TNx | Annual maximum value of daily minimum temperature | |
| | TNn | Annual minimum value of daily minimum temperature | |
| | TX10p | Percentage of days when daily max temperature < 10th percentile | |
| | TX90p | Percentage of days when daily max temperature > 90th percentile | |
| | TN10p | Percentage of days when daily min temperature < 10th percentile | |
| | TN90p | Percentage of days when daily min temperature > 90th percentile | |
| | WSDI | Warm spell duration index: annual count of days with at least 6 consecutive days when TX > 90th percentile | |
| | CSDI | Cold spell duration index: annual count of days with at least 6 consecutive days when TN < 10th percentile | |
| | DTR | Daily temperature range: monthly mean difference between daily max and min temperature | |
| Seasonal or annual mean indices | Tav | Annual average temperature | |
| | SpT | Seasonal average temperature in spring | |
| | SuT | Seasonal average temperature in summer | |
| | AuT | Seasonal average temperature in autumn | |
| | WiT | Seasonal average temperature in winter | |
| ETCCDI extreme indices | R1mm | Annual count of days when precipitation ≥ 1 mm | |
| | R10mm | Annual count of days when precipitation ≥ 10 mm | |
| | R20mm | Annual count of days when precipitation ≥ 20 mm | |
| | CDD | Consecutive dry days: maximum number of consecutive days with daily precipitation < 1 mm | |
| | CWD | Consecutive wet days: maximum number of consecutive days with daily precipitation ≥ 1 mm | |
| | Rx1day | Annual maximum 1-day precipitation | % |
| | Rx5day | Annual maximum consecutive 5-day precipitation | % |
| | SDII | Simple precipitation intensity index | % |
| | R95pTOT | Annual total precipitation when daily precipitation > 95th quantile | % |
| | R99pTOT | Annual total precipitation when daily precipitation > 99th quantile | % |
| Seasonal or annual mean indices | ARav | Annual total precipitation | % |
| | SpR | Seasonal total precipitation in spring | % |
| | SuR | Seasonal total precipitation in summer | % |
| | AuR | Seasonal total precipitation in autumn | % |
| | WiR | Seasonal total precipitation in winter | % |

## 3 Methodology

### 3.1 Subset selection of GCM simulations

Two automated envelope-based methods were used to select subsets of climate simulations. One is the $K$ means clustering which finds cluster centroids that best characterize high-density regions of a multivariate space; the other is the KKZ method which recursively selects simulations that best span the spread of an ensemble (Cannon, 2015). Both selection methods operate on multivariate data, which means that they are sensitive to the choice and scaling of climate variables.

### 3.1.1 Climate variables

Since the hydrological response of a watershed not only depends on annual mean temperature and precipitation but is also sensitive to intra-annual climate variability (e.g. seasonal means or extremes), subset selection should be based on a set of climate variables that includes annual and seasonal averages as well as extremes. The World Meteorological Organization's Expert Team on Climate Change Detection and Indices (ETCCDI) has recommended a set of core climate indices that can be easily derived from daily meteorological data series (http://etccdi.pacificclimate.org/list_27_indices.shtml)TS4. The ETCCDI indices are designed to monitor changes in the frequency and intensity of climate extreme events and characterize the variability of extremes (Zhang et al., 2011). Here, we assume that the ETCCDI indices are sufficient to characterize climate extremes that lead to hydrological impacts.

Specifically, this study used a set of 31 climate variables as shown in Table 2 (21 ETCCDI extreme indices and 10 seasonal or annual mean indices), including 16 temperature variables and 15 precipitation variables. Since the focus of this study is on the capability of selected GCM subsets to cover uncertainty of climate change signals, changes in climate variables (relative change for precipitation and absolute change for temperature and duration) between the historical reference period (1975–2004) and the future projection period (2070–2099) were calculated for 50 climate simulations over the two study watersheds. Changes in each climate variable were standardized to zero mean and unit standard devi-

ation to eliminate influences from different magnitudes and units between variables. These changes in climate variables are referred to as simulated climate change signals. Once changes were calculated, subsets could be selected based on the multivariate space formed by the climate variables.

### 3.1.2 $K$ means clustering

The $K$ means clustering algorithm divided the ensemble of 50 climate simulations into a user-specified number of clusters based on the objective of minimizing within-cluster sum of squared error (SSE) (Hartigan and Wong, 1979). Each cluster is represented by its centroid. The SSE is characterized by the Euclidean distances from simulations to their corresponding cluster centroids. Some studies have applied this method to select subsets of climate simulations (Logan et al., 2011; Cannon, 2015; Houle et al., 2012). The climate simulations closest to the centroids were selected as the subsets. Due to sensitivity of the $K$ means clustering to initial cluster centroid positions, it was run 10 000 times with different initializations and the best solution with lowest SSE was kept. A disadvantage of the $K$ means clustering is that the selected climate simulations are not ordered. In other words, the optimally selected simulations in a small subset may not be optimal in a larger subset, which means that it is inconvenient for end-users to change the subset size for different applications.

### 3.1.3 KKZ method

The KKZ method was originally designed by Katsavounidis et al. (1994) to identify a set of optimal seed cases as initial centroids in the $K$ means clustering, and was introduced by Cannon (2015) in the selection of climate simulations. This method prefers the peripheral simulations in the multivariate space. The specific procedure is as follows.

1. The climate simulation closest to the centroid of the whole ensemble is selected as the first simulation.

2. The simulation farthest from the first selected simulation is selected as the second representative simulation;

3. Subsequent simulations are selected as follows.

   – The distances from each remaining simulation to every previously selected simulation are calculated.

   – Each remaining simulation is designated with the minimum distance among all distances calculated in step 3(1).

   – The simulation with the largest minimum distance, which is designated in step 3(2), is selected as the next selected simulation.

Compared to the $K$ means clustering, the KKZ method is deterministic and ordered. However, it is more susceptible to selecting outliers than $K$ means clustering. In addition, a random selection, repeated 100 times to minimize the influence of its stochastic nature, was conducted as a baseline to evaluate the $K$ means clustering and KKZ method.

### 3.2 Generation of climate scenarios

GCM outputs are typically on a coarse spatial grid and contain systematic biases that preclude their direct use in hydrological modelling (Mpelasoka and Chiew, 2009; Chen et al., 2011a, b; Minville et al., 2008; Vaze and Teng, 2011). It is thus necessary to bias correct and downscale GCM outputs before running the hydrological model. The main objective of this study is to investigate the transferability of climate simulation uncertainty; hence, there is no need to use a complicated downscaling method. A commonly used change factor method, namely the daily scaling (DS) method proposed by Harrold and Jones (2003), was used in this study. This method assumes that climate change signals simulated by GCMs are credible and can be used to perturb observations to obtain future daily series. The DS method adjusts the observed daily series using the differences in distributions of simulated temperature/precipitation between the future period and the reference period. The specific steps are the following.

1. Distributions (represented by 100 quantiles in this study) of daily temperature and precipitation simulated by GCMs are calculated for both reference and future periods in each calendar month (i.e. January, February, etc.);

2. scaling factors are estimated as the differences (for temperatures) or ratios (for precipitation) in distributions of precipitation or temperature between reference and future periods for each calendar month; and

3. scaling factors are added (for temperatures) or multiplied (for precipitation) to corresponding distributions of observed daily temperature or precipitation for each calendar month.

The use of the DS method preserves the simulated climate change signal. It is based on differences in probability distributions between the reference and future periods, which are only caused by climate change signals. In addition, the consideration of quantile-dependent changes in the precipitation distribution is important in hydrological impact studies, because more runoff is generated in high-intensity precipitation events (Harrold and Jones, 2003; Chiew et al., 2009). The use of 100 quantiles in the DS method is the same as many other studies (Mpelasoka and Chiew, 2009; Chen et al., 2013). Lafon et al. (2013) showed that the empirical quantile mapping method based on 100 quantiles is more accurate than that based on 25, 50, or 75 quantiles. However, temporal sequencing in the future period is assumed to be the same as in the observed data. Changes in, for example, wet/dry spell lengths are not informed by the GCM simulations.

**Table 3.** Definitions of 17 hydrological variables. The final column indicates whether the change in a given variable is expressed in the form of relative difference (CT: change type).

| Category | Index | Description | CT |
|---|---|---|---|
| Water resource indicators (WRIs) | MD | Annual mean flow | % |
| | SpMD | Seasonal mean flow in spring | % |
| | SuMD | Seasonal mean flow in summer | % |
| | AuMD | Seasonal mean flow in autumn | % |
| | WiMD | Seasonal mean flow in winter | % |
| | tCMD | Centre of timing of annual flow | |
| Quantiles of daily flow | Q5 | 5th quantile of daily flow series | % |
| | Q50 | 50th quantile of daily flow series | % |
| | Q95 | 95th quantile of daily flow series | % |
| Indicators of hydrological alteration (IHAs) | Qx1day | Annual mean 1-day maximum flow | % |
| | Qx3day | Annual mean 3-day maximum flow | % |
| | Qx7day | Annual mean 7-day maximum flow | % |
| | tQx | Julian date of annual 1-day maximum | |
| | LPC | Number of low pulses (annual median −25th percentile) in a year | |
| | HPC | Number of high pulses (annual median +25th percentile) in a year | |
| | LPD | Mean duration of low pulses in a year | % |
| | HPD | Mean duration of high pulses in a year | % |

## 3.3 Hydrological response simulation

### 3.3.1 Hydrological modelling

The hydrological impacts were simulated by a six-parameter, lumped, conceptual hydrological model, GR4J-6. The GR4J-6 model consists of the GR4J rainfall–runoff model and the CemaNeige snow accumulation and melt routines (Arsenault et al., 2015). The GR4J model is a reservoir-based four-parameter model developed on the basis of the GR3J model (Edijatno et al., 1999; Perrin et al., 2003). This model routes runoff through a production reservoir, two linear unit hydrographs and a non-linear routing reservoir. Four parameters have to be calibrated for this model. They are maximum capacity of the production reservoir, groundwater exchange coefficient, 1-day-ahead maximum capacity of the routine reservoir and time base of unit hydrograph. In an evaluation of the model, Perrin et al. (2003) found that GR4J outperformed 19 models over a large sample of catchments.

Due to its lack of snow accumulation and snowmelt algorithms, the GR4J model cannot be directly used in snow-related watersheds. Thus, the general snow accounting routine proposed by Valéry et al. (2014), CemaNeige, was added. In this routine, precipitation is divided into rainfall and snowfall depending on the daily range of temperatures, and the updating of snowpack and snowmelt is based on a degree-day approach that is controlled by two free parameters (cold content factor and snowmelt factor). In addition, the Oudin formulation (Oudin et al., 2005) was used to preprocess evapotranspiration for GR4J-6.

### 3.3.2 Hydrological variables

To examine the performance of subset selection in terms of hydrological response uncertainty, this study used a set of 17 hydrological variables based on water resource indicators (WRIs), indicators of hydrologic alteration (IHAs), and quantiles of daily flow series (Table 3). WRIs have been used in many hydrological impact studies to assess streamflow alteration due to natural and anthropogenic climate change (Eum et al., 2017; Shrestha et al., 2014; Chen et al., 2011b). IHAs are used to examine the temporal variation of key streamflow hydrograph components (Eum et al., 2017; Richter et al., 1996; Shrestha et al., 2014). Quantiles of daily flow series have been used to describe the characteristics of flow regimes (Mu et al., 2007; Wilby, 2005).

Similar to climate variables, changes in hydrological variables between the reference (1975–2004) and future (2070–2099) periods were calculated. To remove the influence of systematic biases between the observations and simulations, simulated runoff values instead of gauge observations were used as flow data in the reference period. The first year of each period was used to spin up the hydrological model and was excluded when calculating the hydrological variables. Once the projected changes in hydrological variables were calculated, the uncertainty coverage of subsets could be compared between climate variables and hydrological variables to evaluate the transferability of climate simulation uncertainty.

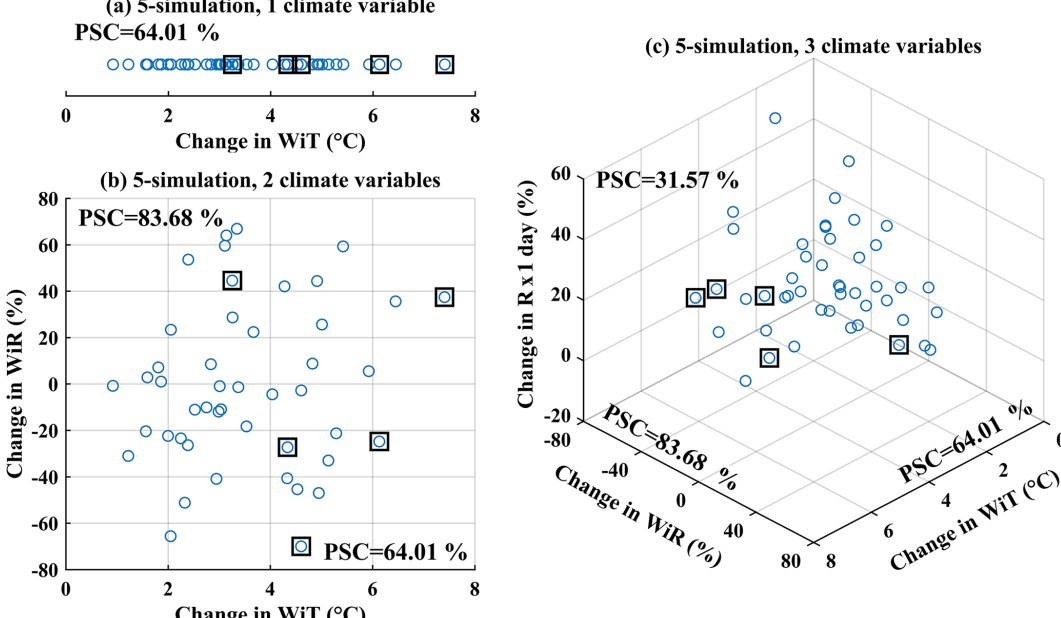

**Figure 2.** Examples of PSCs when selecting five climate simulations over the Xiangjiang watershed using the KKZ method. The PSCs of each variable are presented beside the corresponding axes.

## 3.4 Data analysis

A criterion called the percentage of spread coverage (PSC) is used to measure the uncertainty coverage of a subset relative to the coverage of all simulations. For a given variable and subset, PSC was calculated by dividing the variable's range in the subset by the variable's range in all simulations. A higher PSC means that the selected subset covers a larger uncertainty range. Figure 2 shows examples of PSC when five climate simulations are selected using the KKZ method. Since it is difficult to illustrate results in more than three dimensions, examples are limited to one, two, and three variables. In Fig. 2a, points represent the changes in "WiT" (seasonal average temperature in winter) for 50 GCM simulations. The larger squares represent the same variable for a subset of five climate simulations selected by KKZ. The PSC is calculated by dividing the temperature range of the selected subset, 4.15 °C, by that of the whole ensemble, 6.49 °C. Therefore, for this specific variable the PSC (uncertainty coverage) of the subset is 64.01 %. Similarly, every variable has a corresponding PSC associated with a subset of a given size; examples of "WiR" (seasonal total precipitation in winter) and "Rx1day" (annual maximum 1-day precipitation) are shown in Fig. 2b–c. For the random subset selection method, the reported PSC is the mean value of 100 PSCs, each calculated for a different random subset of the specified size.

## 4 Results

### 4.1 Calibration and validation of hydrological models

The basin-averaged daily minimum and maximum temperatures and precipitation, as shown in Table 4, were used to calibrate and validate the GR4J-6 model over the two watersheds. Model parameters were obtained by the shuffled complex evolution optimization (Duan et al., 1992) based on the objective to maximize Nash–Sutcliffe efficiency (NSE) (Nash and Sutcliffe, 1970). The optimally chosen sets of parameters yield NSE values between 0.87 and 0.93 over both watersheds for calibration and validation. The calibrated GR4J-6 also yields small relative errors of water balance between −0.3 and 5.4 % for calibration and validation. These indicate the reasonable performance of GR4J-6 for both watersheds (Table 4). Figure 3 presents the mean daily hydrographs (average discharge of each calendar day across all years) for the calibration and validation periods over the two watersheds. The calibrated GR4J-6 has good performance in most parts of the year, with the exception of overestimation of winter discharge for the Manicouagan 5 watershed. In addition, since snow accumulation and snowmelt processes are not important in the Xiangjiang watershed, the GR4J model (excluding snow module) was also calibrated in this watershed. Results showed that there was little difference between the calibrated GR4J and GR4J-6 models, and thus the presence of the CemaNeige snow module would not influence the performance of GR4J-6 in the rainfall-characterized Xiangjiang watershed (Table 4).

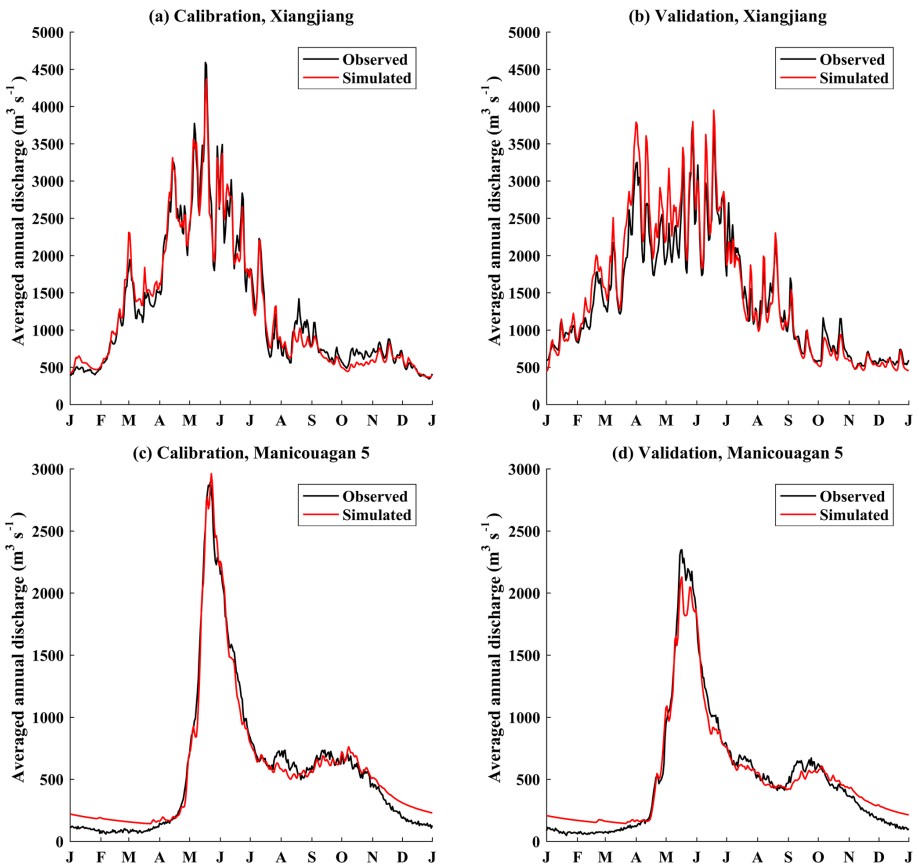

**Figure 3.** Observed and simulated mean hydrographs for **(a, c)** calibration and **(b, d)** validation periods over the **(a, b)** Xiangjiang and **(c, d)** Manicouagan 5 watersheds.

**Table 4.** Nash–Sutcliffe efficiency (NSE) of hydrological models in the calibration and validation over two watersheds.

| Country | Watershed name | Area (km$^2$) | Hydrological model | Calibration period | NSE calibration | RE calibration | Validation period | NSE validation | RE validation |
|---|---|---|---|---|---|---|---|---|---|
| China | Xiangjiang | 52 150 | GR4J-6 | 1975–1987 | 0.912 | −0.3 % | 1988–2000 | 0.871 | 5.4 % |
| | | | GR4J | 1975–1987 | 0.912 | −0.2 % | 1988–2000 | 0.872 | 5.5 % |
| Canada | Manicouagan 5 | 24 610 | GR4J-6 | 1970–1979 | 0.926 | 3.8 % | 1980–1989 | 0.881 | 2.7 % |

## 4.2 Transferability of climate uncertainty

As an illustrative example, the uncertainty transferability from one climate variable to one hydrological variable in the Xiangjiang watershed is shown in Fig. 4. The larger squares represent the 5- and 10-climate simulation subsets selected by the KKZ method. The subfigures on the top display the PSC for "Rx5day" (maximum consecutive 5-day precipitation), whereas those at the bottom display the PSC for "Qx7day" (7-day maximum flow). The reason for choosing these two variables is that there is a generally accepted linkage between high-intensity precipitation and high flow in a rainfall-driven watershed. Although this particular choice of climate and hydrological variables is, in some ways, un-

fair because the overall selection process is based on a high-dimensional multivariate climate space, these subfigures still illustrate the process of uncertainty transferability from climate simulations to hydrological impacts. Here, the PSC of the climate variable increases from 66.45 to 87.37 % as the number of selected simulations goes from 5 to 10; at the same time, the PSC of the hydrological variable increases from 65.42 to 90.32 %. In this case, the uncertainty coverage of the subsets in terms of the climate variable is well translated to uncertainty coverage of the hydrological variable.

Figure 5 expands the example above from one to two dimensions. In this case, the subfigures on the top show a 2-D space formed by the changes in two climate variables, "ARav" (annual total precipitation) and "Rx1day" (maxi-

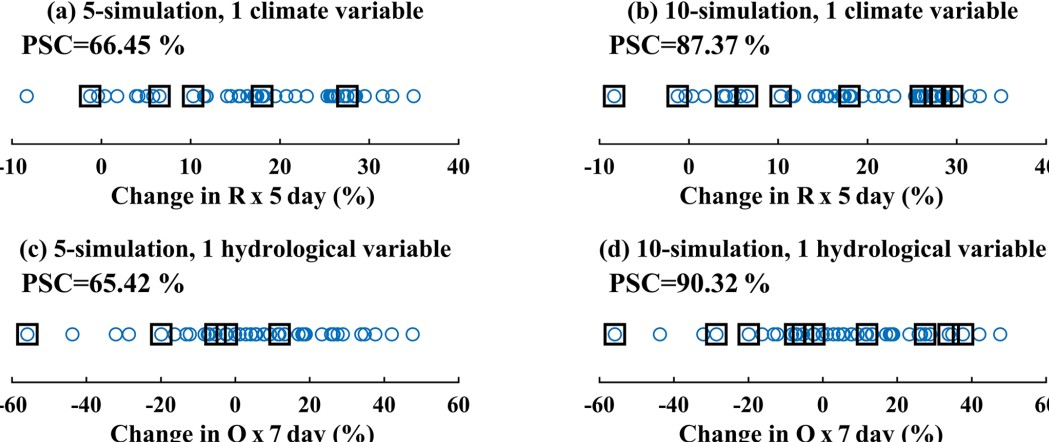

**Figure 4.** Examples of the transferability of climate uncertainty to hydrological impacts based on one variable when selecting **(a, c)** 5 and **(b, d)** 10 climate simulations over the Xiangjiang watershed using the KKZ method. The PSCs of each variable are presented in the top left corner.

mum 1-day precipitation), whereas those at the bottom show changes in two hydrological variables, "MD" (annual mean flow) and "HPD" (mean duration of high pulses). It should be noted that the subsets of climate simulations are the same as in the 1-D example above. As the number of selected simulations increases from 5 to 10, the mean PSC for the two climate variables increases from 50.83 to 88.72 %, while the mean PSC for the two hydrological variables increases from 59.46 to 94.05 %. The increases are mostly due to selection of outlying simulations in the top right corner of the plots (the sixth, eighth, and ninth selected simulations). There is strong consistency between locations of selected simulations in 2-D climate space and hydrology space. For example, the first, fourth, and tenth selected simulations are close to each other in both climate space (Fig. 5b) and hydrology space (Fig. 5d). Accordingly, the uncertainty coverage tends to translate well from climate variables to hydrological variables in this 2-D example. However, PSC increases are not consistent in all cases. For example, selection of the simulation on the left edge of Fig. 5b (the seventh selected simulation) substantially improves the PSC of "Rx1day", but does not lie on the edge of Fig. 5d and hence does not contribute to improvements in PSC of either hydrology variable. This may be due to the non-linearity of the hydrological model or an imperfect explanatory relationship between the climate and hydrological variables.

The discussion above is limited to results for 5- and 10-simulation subsets for one watershed selected using the KKZ method. In the study as a whole, subset sizes from 1 to 50 simulations were evaluated in terms of transferability for two watersheds and two envelope-based methods ($K$ means and KKZ). PSCs for all 31 climate variables and 17 hydrological variables were calculated for both selection methods and watersheds. Figure 6 shows distributions of climate and hydrological PSCs for 5-, 10-, 20-, 30-, and 40-simulation subsets.

For the Xiangjiang watershed (Fig. 6a–b), PSCs for the climate variables are similar to those for the hydrological variables. For the Manicouagan 5 watershed (Fig. 6c–d), PSCs of the hydrological variables are slightly smaller than those for the climate variables. Overall, the tendency of the hydrological PSCs to increase with subset size is comparable to that for the climate PSCs in both watersheds. In other words, as the size of a subset becomes larger, the improvement in PSCs of the hydrological variables is similar to that of the climate variables. When comparing the two envelope-based methods, KKZ tends to outperform $K$ means clustering.

Given the large number of climate and hydrological variables under consideration and the challenges inherent in communicating information about multi-dimensional data, two summary criteria are used to generalize subset coverage results in this study. The first criterion is the average PSC for all climate or hydrological variables. Following Cannon (2015), the second criterion is the percentage of variables that reach a 90 % PSC threshold (PSC90p).

Figure 7 presents the average PSC and PSC90p for climate variables (solid lines) and hydrological variables (dashed lines) when selected subsets contain $K$ simulations ($K = 1$ to 50) over the two watersheds. Generally, the KKZ method performs better than $K$ means clustering for both evaluation criteria and both watersheds, and the two automated envelope-based methods perform better than random selection. In the case of the Xiangjiang watershed (Fig. 7a), the 10-simulation KKZ subset reaches an average PSC of 90 % for climate variables, while $K$ means and random selection require 23 and 29 simulations, respectively, to reach this threshold. For hydrological variables, the KKZ method still shows the best performance. To reach an average PSC of 90 %, KKZ and $K$ means clustering require at least 13 and 26 simulations, respectively. In contrast to results for the climate variables, $K$ means clustering even performs worse than random selection

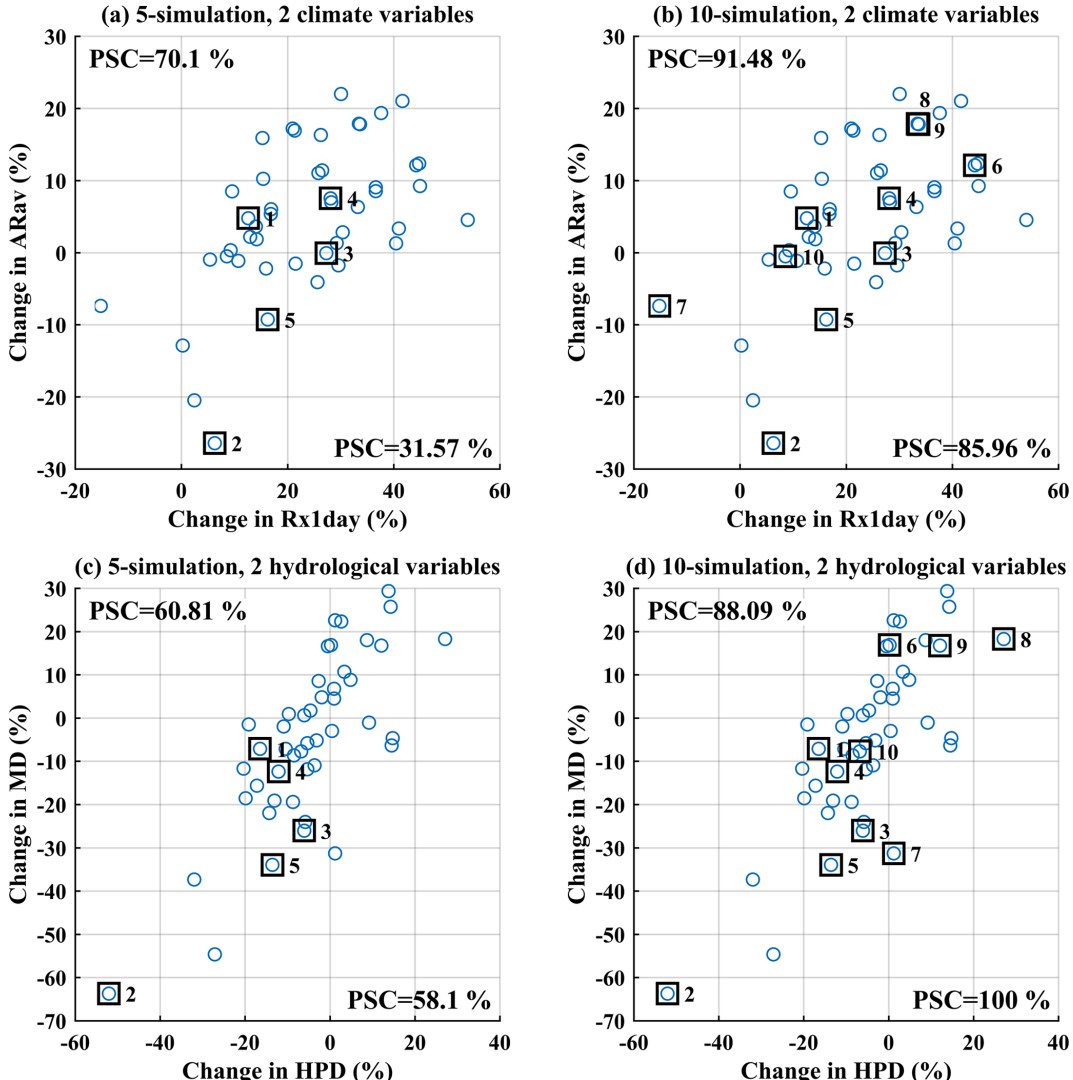

**Figure 5.** Examples of the transferability of climate uncertainty to hydrological impacts based on two variables when selecting **(a, c)** 5 and **(b, d)** 10 climate simulations over the Xiangjiang watershed using the KKZ method. The PSCs of each variable are presented beside the corresponding axes.

for the hydrological variables (e.g. $K = 27$ to 32). In the case of the Manicouagan 5 watershed (Fig. 7c), the KKZ method again outperforms $K$ means clustering and random selection.

In addition, as more simulations are selected, the average PSC increases rapidly when the size of selected simulations is smaller than 10 for both watersheds, while the rate of increase slows when the number is larger than 10. For the KKZ method, a subset of 10 simulations covers more than 85 % of uncertainty for climate and hydrology variables in both watersheds; selecting more than 10 climate simulations leads to little change in uncertainty coverage. For both watersheds, a subset of 10 simulations selected using KKZ appears to be optimal for reducing computational costs while incurring the smallest possible loss of uncertainty information. In addition, the performance of the KKZ method is maintained for larger

subsets, while the performance of $K$ means clustering fluctuates. In other words, a larger subset selected by the $K$ means clustering may not have a greater uncertainty coverage than a smaller subset. The recursive nature of the KKZ method effectively guarantees that average PSC will increase monotonically with subset size.

The focus of this study is the transferability of climate simulation uncertainty to uncertainty in hydrological impacts. For a given method this can be inferred from the difference in average PSC and PSC90p between climate and hydrological variables. For the Xiangjiang watershed, the average PSC of climate variables is close to that of hydrological variables for all selection methods (Fig. 7a). Especially for the KKZ method, differences in average PSC are less than 5 % (with the exception of $K = 2$, 11, and 12). The differences in cli-

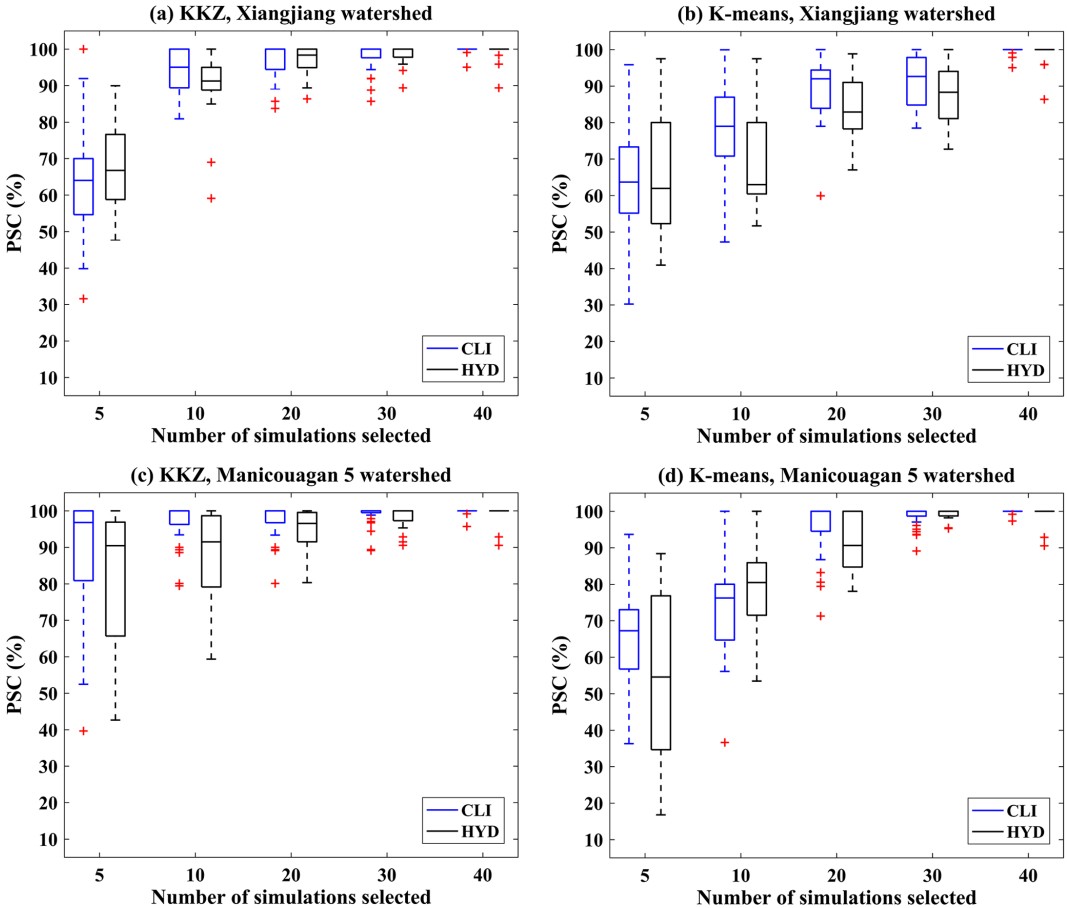

**Figure 6.** Boxplots of the PSCs of 31 climate variables (CLI) and 17 hydrological variables (HYD) when selecting different numbers of climate simulations over two watersheds using the KKZ method and $K$ means clustering.

mate and hydrology uncertainty coverage are slightly larger when using the $K$ means clustering and random selection methods. For PSC90p (Fig. 7b), transferability is somewhat less apparent due to the more rigorous 90 % PSC threshold. Although differences in PSC90p between climate and hydrological variables are sometimes large, especially for the $K$ means clustering, the PSC90p of hydrological variables still exhibits similar overall tendency and behaviour to the climate variables. In general, subsets of climate simulations that are selected based on a large number of relevant climate variables are effective at transferring uncertainty coverage into the realm of hydrological impacts. However, this transferability is method dependent; results are less variable and more consistent for KKZ than $K$ means clustering.

Figure 7c–d present results for average PSC and PSC90p in the Manicouagan 5 watershed. On the whole, the selection methods behave similarly in terms of transferability to in the Xiangjiang watershed, but the uncertainty coverage of the subsets for the hydrological variables is reduced slightly. Degraded transferability is most apparent in larger differences in PSC90p between the climate and hydrological variables.

As noted above, however, this criterion is much more stringent than average PSC.

## 4.3 Impact of temperature variables

The climate variables used in this study can be classified into two groups: temperature variables and precipitation variables. Each variable is given equal weight in the subset selection, regardless of inter-variable correlations, and all variables are assumed to exert the same influence on the hydrological variables. However, the impacts of climate variables on flow regimes may not be the same in watersheds with different hydroclimatic characteristics. For example, warmer temperatures lead to earlier spring floods in northern seasonally snow-covered watersheds (such as the Manicouagan 5 watershed) (Whitfield and Cannon, 2000; Chen et al., 2011b; Minville et al., 2008), whereas changes in temperature have little impact on the timing of floods in rainfall-dominated watersheds (such as the Xiangjiang watershed). Since the importance of temperature is different for the two study watersheds, a question is raised: can the transferability of climate uncertainty in the Xiangjiang watershed be improved

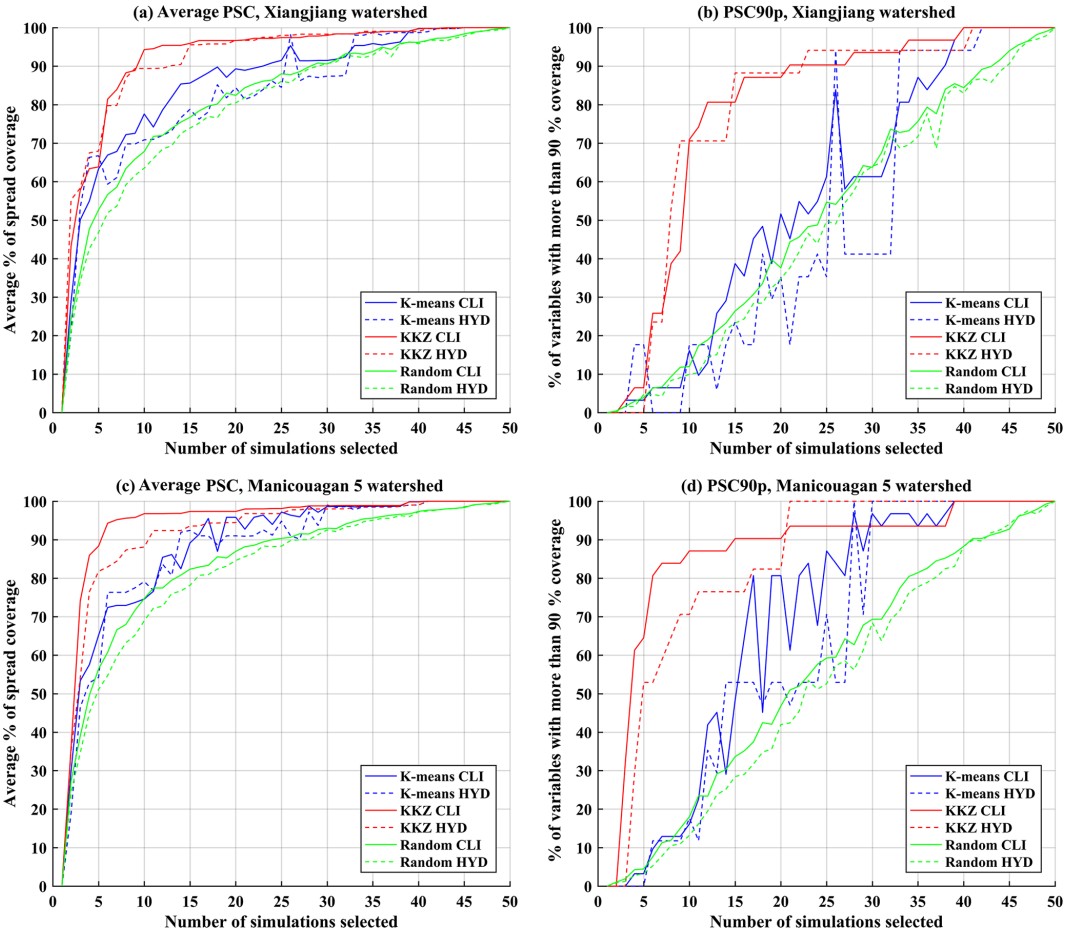

**Figure 7.** The **(a, c)** average PSC and **(b, d)** PSC90p for three different selection methods (*K* means, KKZ, and random selection) over the **(a, b)** Xiangjiang watershed and the **(b, d)** Manicouagan 5 watershed (CLI: climate variables; HYD: hydrological variables).

if irrelevant temperature variables are removed? To answer this question, temperature variables (the first 16 variables in Table 2) were removed and subset selection was conducted again using the 15 precipitation variables. The average PSC and PSC90p were then calculated to compare with original results that include temperature variables. Results from the precipitation analysis are shown in Fig. 8.

For the Xiangjiang watershed (Fig. 8a–b), removing temperature variables from the subset selection leads to improved uncertainty coverage for the hydrological variables, especially for *K* means clustering. The *K* means clustering now performs better than random selection in most cases. For KKZ, average PSC for the hydrological variables reaches 90 % with a subset of only 6 simulations, whereas the same level of coverage required 13 simulations when considering both temperature and precipitation. However, the effect of removing temperature variables is the opposite for the Manicouagan 5 watershed (Fig. 8c–d). Here, coverage performance for the hydrological variables is reduced when temperature variables are not considered. The contrasting effects are consistent with the processes that generate runoff in the

two watersheds. As mentioned above, the Manicouagan 5 watershed is seasonally snow-covered – snow accumulation and snowmelt are the dominant processes that contribute to runoff generation – and hence it is sensitive to changes in temperature. However, temperature variables are not relevant in the rainfall-dominated Xiangjiang watershed. The different impacts of temperature variables in the two watersheds highlight the necessity of carefully choosing climate variables for subset selection based on physical process knowledge.

## 4.4 Transferability of the multi-model mean

In addition to the overall spread in the projected climate change signal, policymakers are also concerned with the MME mean when communicating hydrological climate change impacts. Therefore, the selection methods are also evaluated in terms of their ability to preserve the multi-model mean of the full MME. It bears noting that the CMIP5 MME considered in this study is an ensemble of opportunity. Models are not statistically independent, for example

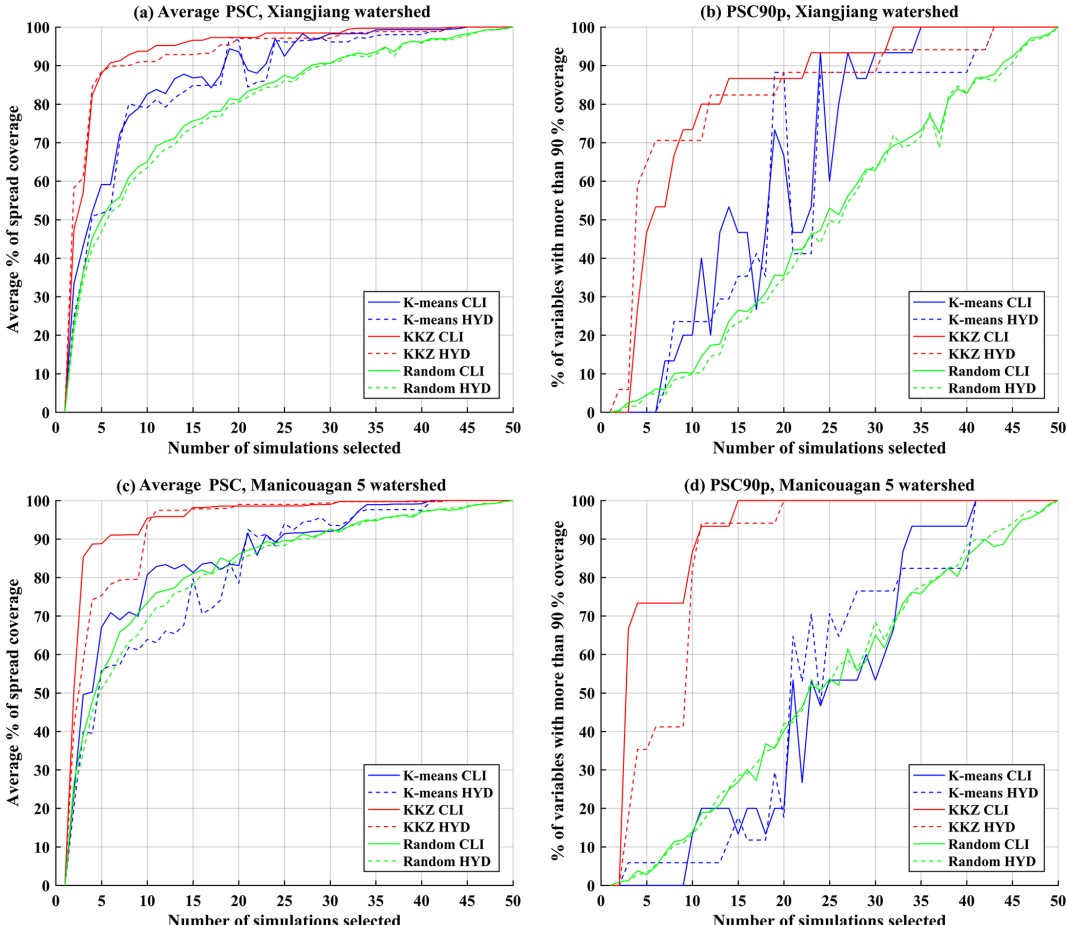

**Figure 8.** The **(a, c)** average PSC and **(b, d)** PSC90p for three different selection methods (*K* means, KKZ, and random selection) over two watersheds when temperature variables are excluded in the process of simulation selection (CLI: climate variables; HYD: hydrological variables).

due to shared physical parameterizations, and multiple simulations may be contributed by the same model. Also, the two envelope-based methods make very different assumptions about the underlying nature of the statistical distribu-
⁵ tion of the ensemble. The KKZ method is not biased towards high-density regions of the multivariate space, preferring uniform coverage, whereas the *K* means method, which assumes a mixture of multivariate normal clusters with equal variance, will tend to select simulations that lie in regions
¹⁰ populated by a large number of simulations. These characteristics will have implications for preservation of the MME mean.

In order to generalize the MME mean over multiple variables, standardized changes in each variable are averaged
¹⁵ across variables and selected simulations to obtain a dimensionless criterion (referred to as averaged standardized mean change). For different sized subsets selected by the three selection methods, corresponding climate and hydrological averaged standardized mean changes were calculated
²⁰ and compared with values for the whole ensemble. Because

projected changes are preprocessed by standardizing to zero mean and unit standard deviation, the averaged standardized mean change in the whole ensemble is zero by construction. Therefore, if the averaged standardized mean change in a subset is close to zero, the MME mean change simu- ²⁵ lated by that subset is similar to that simulated by the entire ensemble. Figure 9 shows the averaged standardized mean changes in climate and hydrological variables when *K* simulations (*K* = 1 to 50) are selected for the two watersheds. When averaged over a large number of random trials, mean ³⁰ values will, by definition, lie close to zero for the random selection method; thus, the envelopes of results across all 100 random selections are presented as blue and pink shaded areas in each subfigure for climate and hydrological variables, respectively. Figure 9a–b present results for subsets when ³⁵ temperature variables are included in the selection process, whereas Fig. 9c–d present results when temperature variables are excluded.

Overall, when gauged against the range of variability in the 100 random selections, subsets selected by both statis- ⁴⁰

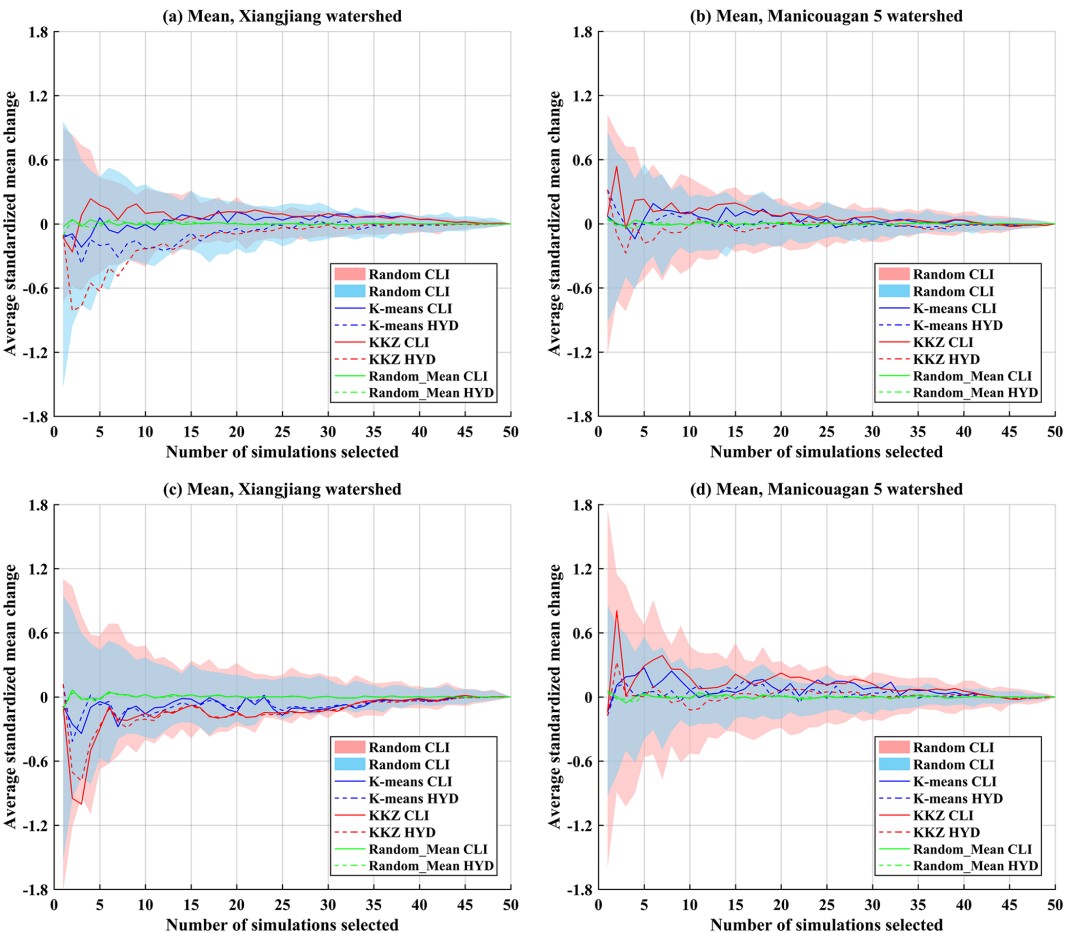

**Figure 9.** Averaged standardized mean changes in climate (CLI) and hydrological (HYD) variables of subsets selected by three selection methods ($K$ means, KKZ and random selection) over the Xiangjiang and Manicouagan 5 watersheds when temperature variables are **(a, b)** included or **(c, d)** excluded in the process of selection. The pink and blue panels are the envelopes resulting from 100 random selections.

tical methods perform well in reproducing the MME mean of the entire ensemble, with $K$ means clustering performing slightly better than the KKZ method. When looked at in more detail, in the Xiangjiang watershed, the averaged standardized mean changes of subsets in climate variables tend to differ slightly from those in hydrological variables when temperature variables are included (Fig. 9a). For example, when five simulations are selected using the KKZ method, the averaged standardized mean change for climate variables is 0.18, whereas it is $-0.63$ for hydrological variables. Subsets selected by the KKZ method often have higher means than the whole ensemble for climate variables, while they have lower values for hydrological variables. In other words, a subset with positive changes in climate variables gives negative changes in hydrological variables, which means that selected subsets have poor transferability in terms of MME mean. However, when temperature variables are not included in the selection process, the transferability of the multi-model mean is improved (Fig. 9c). In the Manicouagan 5 watershed, by contrast, differences between average changes in cli-

mate variables and hydrological variables are slightly smaller when temperature variables are included (Fig. 9b, d). Again, this highlights the importance of selecting the appropriate climate variables when performing ensemble subset selection.

## 5 Discussion

In order to recommend a practical subset of climate simulations for end-users who deal with the assessment of climate change impacts on hydrology, various selection methods have been proposed based on different criteria (Mendlik and Gobiet, 2016; Cannon, 2015; Gleckler et al., 2008; Lutz et al., 2016a; McSweeney et al., 2012; Warszawski et al., 2014; Perkins et al., 2007). Even though these methods usually perform well in terms of the climate variables to which they are applied, their performance in terms of hydrological impacts needs to be verified. In normal usage, for example, envelope-based methods may only consider changes in mean temperature and annual precipitation (Immerzeel et al., 2013; Murdock and Spittlehouse, 2011; Warszawski et al., 2014),

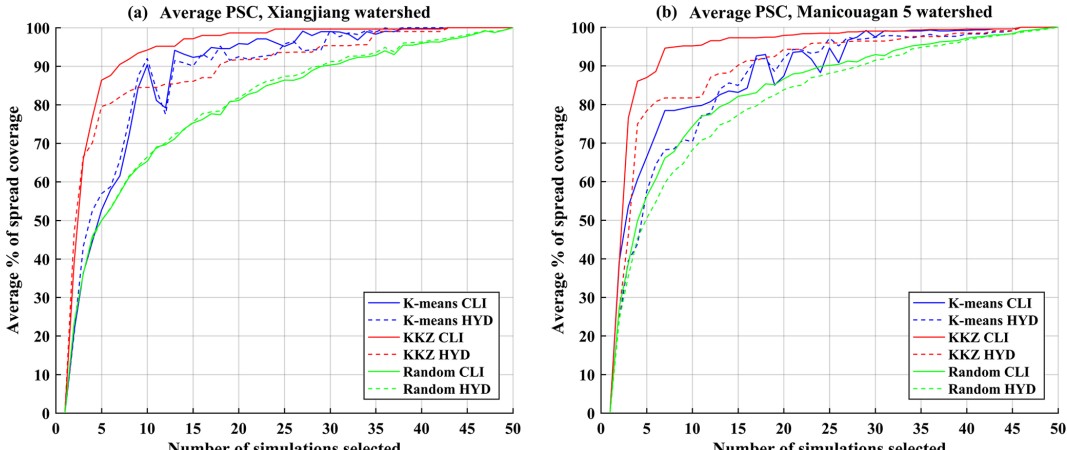

**Figure 10.** The average PSC for three different selection methods (*K* means, KKZ and random selection) over two watersheds when using QM methods (temperature variables are excluded in the Xiangjiang watershed, while they are included in the Manicouagan 5 watershed).

which will have a strong influence on both the overall measurement of climate uncertainty and subset selection results in terms of hydrological impacts. By not considering relevant climate variables, there may be a loss of information when transferring climate uncertainty to hydrological uncertainty (Chen et al., 2016). When one considers the fact that it is often hard to determine a one-to-one correspondence between climate and hydrological variables, it may be reasonable to use a large suite of climate variables.

Therefore, this study investigated the transferability of climate simulation uncertainty to the hydrological world by *K* means clustering and KKZ methods using a large number of climate and hydrological variables, including both seasonal and annual means and extremes. Multiple variables, when selected carefully, can improve the transferability of climate simulation uncertainty to hydrology impacts. Although the introduction of multiple climate variables may lead to redundant information, and it may be unnecessary for impact studies with different aims (e.g. water balance or hydrological drought) to consider so many climate extreme indices, this general approach can nonetheless give a more useful and reasonable selection for the purpose of covering an overall range of future climate change and its hydrological impacts. This is crucial for hydrological modellers as they usually spend a lot of computational costs in running a large number of climate simulations with a complicated hydrological model. The results of this study show that the climate simulation uncertainty is transferable in the envelope-based selection based on multiple climate variables, and the subset of around 10 climate simulations can cover the majority of uncertainty. Thus, the selected 10 climate simulations can be directly used to drive a hydrological model for impact studies instead of using all climate simulations. In addition, depending on the choice of climate variables and climate model ensemble, it may not be necessary to extract, store, and compute climate indices from all climate model simulations in the en-

semble of interest. For example, pre-computed ETCCDI climate extreme indices for GCMs in CMIP3 and CMIP5 are publicly available from http://climate-modelling.canada.ca/climatemodeldata/climdex/. TS5

In order to further investigate the impact of variable selection on transferable uncertainty, this study also compares uncertainty transferability in a rainfall-dominated watershed and a seasonally snow-covered watershed when including and excluding temperature variables. The different impacts of temperature variables over two watersheds indicate that climate variables, if not chosen with consideration of runoff-generating processes, can affect the performance of the subset selection algorithms. In the rainfall-dominated Xiangjiang watershed, inclusion of temperature variables, which play little role in generating runoff, leads to a small loss of performance, whereas in the snow-related Manicouagan 5 watershed, exclusion of temperature variables results in a large loss of performance. This is reflected in results for both ensemble spread and MME mean. Thus, it is important to choose proper climate variables that characterize the physical processes controlling the hydrology of the watershed for subset selection. Both watersheds in this study are located in humid regions with moderately hilly topography. The proper climate variables for watersheds in other climate regions may need to be modified accordingly. For example, for high-mountain regions, precipitation may be influenced by complex topography and snowmelt often has the greatest contribution to runoff, and thus it is recommended that climate variables related to orographic precipitation and the evolution of snowpack be included in the selection process (Cannon et al., 2017; Immerzeel et al., 2012); for arid regions, Hortonian overland flow may be the predominant runoff mechanism, and thus precipitation variables that are capable of describing the intensity of precipitation events may need to be stressed (Pilgrim et al., 1988). End-users can also choose the group of climate variables according to their

knowledge of the climate and hydrological characteristics of the watershed of interest and then select the representative subset of climate simulations to save computational costs in the hydrology world, although it is likely that some level of site and study-specific analysis will be required in other climate regions. However, the judgement on relevant climate variables in this study is somewhat subjective. Some automated variable selection procedure may provide a more objective selection of relevant climate variables, such as redundancy analysis or multivariate sparse group lasso (Li et al., 2015).

In terms of selection methods, the results of this study reveal two strengths of the KKZ method over $K$ means clustering. First, the KKZ method selects simulations on the boundaries of the climate simulation ensemble and, as a result, it is better able to cover overall climate uncertainty, as measured by average PSC and PSC90p, of the ensemble than $K$ means clustering. Second, uncertainty coverage of the KKZ method for climate variables increases monotonically as more climate simulations are selected, whereas the $K$ means clustering is unstable. This is because climate simulations are added incrementally, in a recursive fashion, by the KKZ method as subset size increases, whereas $K$ means clustering needs to be run independently for each subset. Consequently, $K$ means clustering produces a disordered sequence of solutions. The results of this study show that these two strengths of the KKZ method are retained for hydrological impacts. Therefore, in the aspect of overall uncertainty coverage, the KKZ method outperforms $K$ means clustering. Performance in terms of MME mean was also evaluated in this study. Results show that the subsets selected by $K$ means clustering produce a more similar MME mean to the whole ensemble, although differences between the two methods are small. This result is expected because $K$ means clustering selects representative simulations for each cluster according to their closeness to the cluster centroid, which is the multivariate mean.

The two envelope-based methods in this study are from a single branch of selection methods whose purpose is to cover the spread (uncertainty) in projected changes of an ensemble. The model ranking approach is another common way to select model simulations, usually based on historical model performance, measures of statistical independence, and other evaluation metrics. Some studies have investigated the impact of weighting GCMs on the projection of climate conditions or hydrological impacts (Chen et al., 2017; Christensen et al., 2010). They concluded that weighting methods have little influence on the ensemble mean and uncertainty, and it is more appropriate to consider GCMs as being equiprobable.

Some studies have argued that certain GCMs may not be independent of one another because of shared code or parameterization schemes (Evans et al., 2013; Knutti et al., 2010; Mendlik and Gobiet, 2016). In an ensemble of opportunity like CMIP5, this dependence may lead to high-density regions of climate variable space and hence influence the selection of models by methods like $K$ means clustering. On the other hand, the KKZ method is designed to select simulations that lie on the edges of the ensemble. If these simulations are outliers because their projections are not credible, for example due to poor process representation, then their selection may not be warranted. Therefore, previously removing any obviously dependent or ill-behaving GCMs through model weighting methods may improve the rationality of these two equal-weighting selection methods in regional impact studies.

In this study, only one downscaling method was used to generate climate scenarios in the scale of a watershed. In order to consider different downscaling methods, the quantile mapping (QM) approach (Maurer and Pierce, 2014; Piani et al., 2010) was additionally examined. Figure 10 presents the results of average PSC in this case. Compared with the results of the DS method, the overall character of the results is roughly the same. Thus, the choice of a downscaling method may have little influence on the conclusions of this study. In addition, only two emission scenarios (RCP4.5 and RCP8.5) were considered. RCP4.5 is the medium stabilization scenario and RCP8.5 represents the very high radiative forcing scenario. The mitigation scenario, RCP2.6, was not used because recent analyses suggested that this RCP will be very difficult to achieve with current emission trajectories (Arora et al., 2011; Rozenberg et al., 2015). RCP6.0 is a scenario with radiative forcing that is bracketed by RCP4.5 and RCP8.5 and was not simulated by as many modelling centres as RCP4.5 and RCP8.5. Thus, RCP4.5 and RCP8.5 were used to include a range of realistic projections (Lutz et al., 2016b). Due to unknown future emission scenarios, two concentration pathways were used in an undifferentiated manner to cover uncertainty resulting from emission scenarios. However, the two emission scenarios do generate different climate simulations. Thus, each pathway was also separately input into the subset selection to examine the transferability when only one scenario was considered. Figure 11 shows the average PSC for either emission scenario over the Xiangjiang watershed. The main character of the results is the same as the original research, where two RCPs were considered and selection of five or six climate simulations by the KKZ method can cover an adequate uncertainty range. Therefore, the specific choice of emission scenarios can be decided by end-users according to their own needs.

## 6  Conclusions

In this study, the transferability of climate simulation uncertainty to climate change impacts on hydrology was investigated over two watersheds with different climate and hydrological regimes based on multiple climate variables. The main conclusions are summarized as follows.

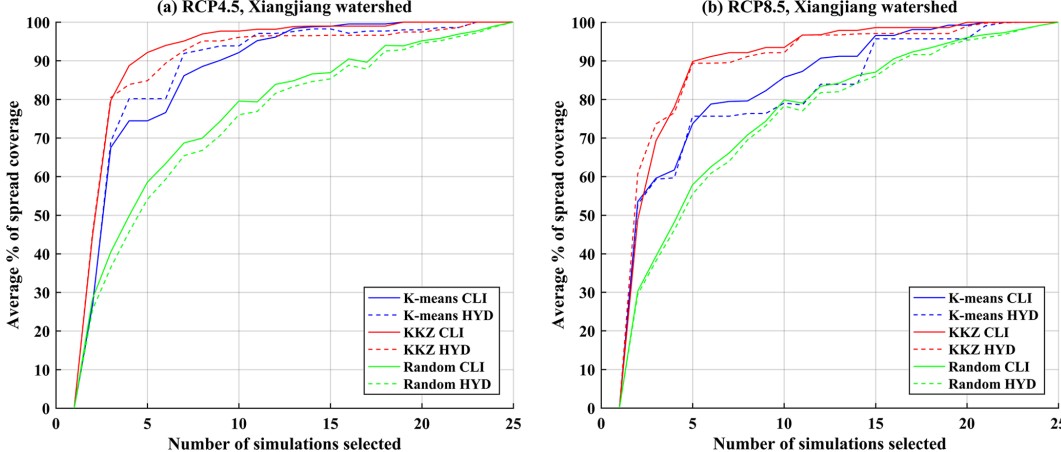

**Figure 11.** The average PSC for three different selection methods ($K$ means, KKZ and random selection) over the Xiangjiang watershed when only one emission scenario (RCP4.5 or RCP8.5) is considered.

1. In terms of uncertainty coverage, both the KKZ method and $K$ means clustering are effective at selecting subsets of climate simulations that represent the range of the climate change signal. However, when it comes to hydrological impacts, the KKZ method always performed better than random selection, while $K$ means clustering sometimes performed worse than random selection.

2. Both $K$ means clustering and the KKZ method are capable of reproducing the MME mean of the whole ensemble, although $K$ means clustering performed slightly better than the KKZ method in some cases.

3. The uncertainty of climate simulations based on multiple climate variables can be transferred to the assessment of hydrological impact uncertainty. In other words, selected subsets can generate similar uncertainty coverage in terms of both climate simulation and hydrological impacts.

4. In order to cover an adequate range of climate simulation and hydrological impact uncertainty with fewer computational costs, selection of about 10 simulations from the ensemble of 50 simulations is required. Little improvement is gained when the number of simulations is increased beyond 10.

5. The choice of climate variables affects the transferability of climate uncertainty to hydrological uncertainty. Thus, the climate and hydrological regimes of a watershed should be considered when choosing variables used to subset climate model simulations for hydrological impact studies.

*Data availability.* . TS6

*Competing interests.* The authors declare that they have no conflict of interest. TS7

*Acknowledgements.* This work was partially supported by the National Natural Science Foundation of China (grant nos. 51779176, 51339004, and 51539009), the Thousand Youth Talents Plan from the Organization Department of CCP Central Committee (Wuhan University, China), and the Research Council of Norway (FRINATEK Project 274310). The authors would like to acknowledge the contribution of the World Climate Research Program Working Group on Coupled Modelling, and all climate modelling groups listed in Table 1 for making available their respective model outputs. The authors would also like to acknowledge Hydro-Québec and the Changjiang Water Resources Commission for providing observation data in the Manicouagan 5 and Xiangjiang watersheds, respectively.

Edited by: Kerstin Stahl
Reviewed by: two anonymous referees

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

TS3 Please add last access.

TS4 Please add last access.

TS5 Please add last access.

TS6 Please provide a statement on how your underlying research data can be accessed. If the data are not publicly accessible, a detailed explanation of why this is the case is required. The best way to provide access to data is by depositing them (as well as related metadata) in reliable public data repositories, assigning digital object identifiers (DOIs), and properly citing data sets as individual contributions. Please indicate if different data sets are deposited in different repositories or if data from a third party were used. If no DOI is available, assets can be linked through persistent URLs to the data set itself (not to the repositories' home page). This is not seen as best practice and the persistence of the URL must be secured.

TS7 Declaration of all potential conflicts of interest is required by us as this is an integral aspect of a transparent record of scientific work. If there are possible conflicts of interest, please state what competing interests are relevant to your work.