# Peer review of "Transferability of climate simulation uncertainty to hydrological impacts"

_Hydrology and Earth System Sciences, 2017_

## Referee Comment (RC1) · Anonymous Referee #1 · 7 Feb 2018

The manuscript is about the transferability of the climate model uncertainties, introduced by the selection of climate models, to hydrological impacts. To this end, two envelope-based selection methods, K-means clustering and the Katsavounidis-Kuo-Zhang (KKZ) method, are used to select subsets from an ensemble of 50 climate models over two watersheds with different climate characteristics. The transferability of the climate model uncertainties is evaluated by comparing uncertainty coverage between 31 climate variables and 16 hydrological variables that are simulated by the hydrological model GR4J. In addition, also the importance of choosing climate variables properly while selecting subsets is investigated by in- and excluding temperature variables. The manuscript is well structured and written. The manuscript covers a topic that is original and novel, and might interest a large amount of readers, including climate scientists

and hydrologists. To my opinion, this manuscript needs some minor revisions. I have added a few comments/suggestions that need to be addressed before acceptance.

General Comments

I would rephrase the title a bit. In the title, the authors are referring to the transferability of climate simulation (model?) uncertainty to "hydrological climate change impacts", whereas in the Abstract and other parts of the manuscript the authors write about the transferability to "hydrological impacts". I would change the "hydrological climate change impact" into "hydrological impacts". In this way, the authors can put more emphasis on the transferability of uncertainties to "hydrological impacts" specifically, and the title has a better connection with the Abstract and manuscript or vice versa.

Two watersheds with different climate characteristics are selected for this study. It would be good to spent some text in the Discussion explaining what the potential effects of transferability are in other climatic regions, such as high-mountain regions.

Specific Comments

1. Introduction; L4-9: the authors indicate that the selection methods inherit the potential flaws of the past-performance approach, when the emphasis is on model performance. What are the potential flaws of the past-performance approach? Combining envelope coverage criteria and past-performance would, to my opinion, be better since not only the models are selected to represent a full range of climate conditions, but also are tested in their performance to simulate regional (historical) climate characteristics, especially in those regions where, for instance, monsoon systems prevail.

1. Introduction; L20-25: To my opinion, the number of variables that is chosen for a selection approach depend on what the scope is of the study. If a study has only a focus on projecting changes in water availability or changes in the water balance it would, to my opinion, not be necessary to take indices into account that represent climatic extremes, whereas a study with a focus on hydrological extremes needs to include

these indices in the selection approach. Therefore, the authors need to elaborate more on why a certain number of variables should be selected or not.

2.2.1 Climate Simulations; L9: Why did the authors select 50 models? The authors mentioned before that the CMIP5 archive includes 61 models. Is there a reason why the other 11 models are excluded from the selection approaches? In addition, each climate model has one or more ensemble member. Did the authors select the first ensemble member or did they select random ensemble members? The authors need to include this information in the method description, for instance by adding extra information to Table 1.

2.2.2 Observations; L17: Where are the data from the 100 rain gauges, 8 temperature gauges, and 1 streamflow gauge obtained? The authors have to include some references to the sources where they obtained the meteorological and discharge data.

3.2 Generation of Climate Scenarios: Why did the authors use the DS method to downscale GCMs and not a method such as the Advanced Delta Change approach or the Quantile Mapping approach that also take changes in extremes into consideration? It might be interesting to discuss potential uncertainties that are introduced by the downscaling approaches in the Discussion.

3.3.1 Hydrological Modelling; L15-21: I would recommend replacing and to discuss this part in more detail in the Results Section, for instance under a separate subsection "Calibration and Validation"

3.4 Data Analysis: It would be good to an additional sentence on what it means when a PSC is high or low. For the reader, it might be more difficult to image the meaning of a high or low PSC.

Figure 1: The longitude axes are given, but the latitude axes are missing. Further I think the figure does not contain a lot of information. I would add a digital elevation model or another topographic/geographic info to give the reader more valuable information

on the characteristics of the catchments. In addition, I would add the positions of the discharge gauging stations used for the calibration/validation. Finally, I recommend making inlets, including the catchment maps, larger.

Technical Comments

Abstract; L16: "...the importance of choosing climate variables properly while selecting subsets..." instead of "...the importance of properly choosing climate variables in selecting subsets..."

---

## Referee Comment (RC2) · Anonymous Referee #2 · 12 Feb 2018

The authors do a good job in their attempt to shed light on the important problem that impact modelers face in efficiently and effectively capturing the range of uncertainty in climate model simulations. Furthermore, they investigate whether covering this range in climate model output variables translates to capturing the uncertainty range of hydrological variables. The paper is well written and clearly presented. Though, in the end, I was not convinced that impact modelers can actually save much time and effort using this methodology. I would recommend that the manuscript needs minor revisions. Importantly, the authors need to make it clearer how an end user can avoid downloading all 50 simulations in order to prove which subset of 10 are most appropriate to cover the uncertainty range in their study.

I would begin by asking this. What do end users or impact modelers gain by this paper? You have shown that for your two different watersheds, a subset of approximately 10 model simulations are needed to reasonably capture the spread of the model uncertainty for both climate and hydrological variables. Additionally, you point out that not using the temperature variables to obtain the subset affects the hydrology of the two watersheds differently. As a result, you illustrate that the selection of the 10 climate models is unique to each impact assessment study. Furthermore, you needed all 50 simulations to test which subset was optimal for your two cases. I do not see how an impact modeler would not need to repeat precisely what you have done. In order to replicate your method, but specific to their study interest or area, they would need to "extract, store, and compute" (page 2, line 14) all 50 model simulations themselves. Then, couldn't they just as easily implement the entire set of simulations instead of a smaller subset? To ask it more directly: How can an impact modeler know which 10 model simulations to use, for their unique case, without testing the ensemble ranges of each possible subset with respect to the entire set of simulations? And to do this, would they not need to run all 50 model simulations?

Some more specific comments and questions are as follows:

In section "2.2.1 Climate Simulations": Does it make sense to lump the uncertainty ranges of both RCP4.5 and RCP8.5? These are two different concentration pathways that represent very different conditions. It is true that we currently can't know which is more likely. I would recommend either treating each pathway independently with different ranges of uncertainty, or I would recommend also including simulations from pathways RCP2.6 and RCP6.

Page 7, line 17: What was the reason to use 100 quantiles instead of the total number of days in the periods (e.g., 1975-2004 for January = 30 years times 31 days = 930 days or quantiles)?

Page 9, line 5: I do not anticipate for it to change your results that much, but perhaps it is better to use something like standard deviation as a measure of the uncertainty

coverage. The Percentage of Spread Coverage (PSC) is only sensitive to the range of the minimum and maximum values. You could end up having many of the models grouped close together, and as a result, your measure would overestimate your actual uncertainty coverage.

Figure 2: Are you showing the observed and simulated values for the calibration and validation for 1 year? Or is each day the average of that day across the years (e.g., for Xiangjiang: all January 1 values are averaged over the time period 1975-1987, then January 2 values are averaged over the same years, . . .)?

---

## Author Comment (AC1) · 29 Mar 2018

Dear Anonymous Referee #1,

We sincerely appreciate the referee's comments and suggestions on the manuscript. All suggestions are helpful to improve this manuscript. We have carefully studied, considered and responded to all comments point-by-point. Please check the attached replies. The manuscript will be modified accordingly.

Please also note the supplement to this comment: https://www.hydrol-earth-syst-sci-discuss.net/hess-2017-703/hess-2017-703-AC1-supplement.pdf

[Figure]

**Supplement:**

Replies to Referee #1

**Transferability of climate simulation uncertainty to hydrological climate change impacts**

Hui-Min Wang, Jie Chen, Alex J. Cannon, Chong-Yu Xu, Hua Chen

We sincerely appreciate the referee's comments and suggestions on the manuscript. All suggestions are helpful to improve this manuscript. We have carefully studied, considered and responded to all comments point-by-point as follows. For clarity, all comments are given in italics and responses are given in plain text. The manuscript will be modified accordingly.

> *The manuscript is about the transferability of the climate model uncertainties, introduced by the selection of climate models, to hydrological impacts. To this end, two envelope-based selection methods, K-means clustering and the Katsavounidis-Kuo-Zhang (KKZ) method, are used to select subsets from an ensemble of 50 climate models over two watersheds with different climate characteristics. The transferability of the climate model uncertainties is evaluated by comparing uncertainty coverage between 31 climate variables and 16 hydrological variables that are simulated by the hydrological model GR4J. In addition, also the importance of choosing climate variables properly while selecting subsets is investigated by in- and excluding temperature variables. The manuscript is well structured and written. The manuscript covers a topic that is original and novel, and might interest a large amount of readers, including climate scientists and hydrologists. To my opinion, this manuscript needs some minor revisions. I have added a few comments/suggestions that need to be addressed before acceptance.*

We appreciate that the reviewer is in favor of the content of this research. All the comments and suggestions have been replied to below and will be addressed in the revision.

> *General Comments*

> *I would rephrase the title a bit. In the title, the authors are referring to the transferability of climate simulation (model?) uncertainty to "hydrological climate change impacts", whereas in the Abstract and other parts of the manuscript the authors write about the transferability to "hydrological impacts". I would change the "hydrological climate change impact" into "hydrological impacts". In this way, the authors can put more emphasis on the transferability of uncertainties to "hydrological impacts" specifically, and the title has a better connection with the Abstract and manuscript or vice versa.*

Thank you for the suggestion on the title. The "hydrological climate change impacts" will be changed to "hydrological impacts" in the revised manuscript. With regard to the use of "climate

simulation" instead of "climate model", although global climate models (GCMs) are the main uncertainty source considered, this research also includes uncertainty related to future emission scenarios (i.e. RCP4.5 and RCP8.5). Thus, we think that "climate simulation" is more appropriate than "climate model" in the title.

Following reviewer's suggestion, the title will be changed to "Transferability of climate simulation uncertainty to hydrological impacts" in the revised manuscript.

> *Two watersheds with different climate characteristics are selected for this study. It would be good to spend some text in the Discussion explaining what the potential effects of transferability are in other climatic regions, such as high-mountain regions.*

Thanks for the comment. We agree with the reviewer that it is necessary to further discuss the transferability in other climatic regions.

For example, for high-mountain regions, precipitation may be influenced by complex topography and snowmelt often has the greatest contribution to runoff. Therefore, it is recommended that climate variables related to orographic precipitation and the evolution of snowpack be included in the selection process. Additionally, for arid regions, Hortonian overland flow may be the predominant runoff mechanism, and thus precipitation variables that are capable of describing the intensity of precipitation events may need to be stressed in the selection of subsets.

These points will be added to the Discussion section of the revised manuscript.

> ### *Specific Comments*
>
> *1. Introduction; L4-9: the authors indicate that the selection methods inherit the potential flaws of the past-performance approach, when the emphasis is on model performance. What are the potential flaws of the past-performance approach? Combining envelope coverage criteria and past-performance would, to my opinion, be better since not only the models are selected to represent a full range of climate conditions, but also are tested in their performance to simulate regional (historical) climate characteristics, especially in those regions where, for instance, monsoon systems prevail.*

Thank you for the comment. In this sentence, the potential flaws are meant as "In general, the assumption that models with good performance over the near-past provide more realistic climate change signals is questionable (Knutti et al., 2010; Reifen and Toumi, 2009), and the metrics commonly used to evaluate model performance are often manually defined based on the fields of interest, which leads to substantial subjectivity within the weighting process (Mendlik and Gobiet, 2016)". Specifically, the best performing models may not produce the most realistic climate change signal in the future. In addition, the ranking or weighting GCMs is highly dependent on the definition of metrics. Thus, the past-performance approach may lead to subjectivity in the selection

of climate model simulations.

With regards to combining envelope coverage criteria and past-performance criteria, on one hand, we actually wanted to say (but failed to say it clear enough) that many approaches combining both criteria put more emphasis on the past-performance criteria. These methods may inherit the potential flaws of the past-performance approach. On the other hand, we agree with the reviewer that combining envelope coverage criteria and past-performance criteria may be better at selecting climate simulations for impact studies. In other words, it may be more reasonable to remove unrealistic models rather than keep models with "best performance". This point has been stated in the 3rd paragraph of the Introduction [Page 2, Lines 26-27] and in the last paragraph of the Discussion section [Pages 15, Lines 9-11]. This question deserves further investigation.

All related information will be clarified in the revised manuscript.

> *1. Introduction; L20-25: To my opinion, the number of variables that is chosen for a selection approach depend on what the scope is of the study. If a study has only a focus on projecting changes in water availability or changes in the water balance it would, to my opinion, not be necessary to take indices into account that represent climatic extremes, whereas a study with a focus on hydrological extremes needs to include these indices in the selection approach. Therefore, the authors need to elaborate more on why a certain number of variables should be selected or not.*

Thanks for this comment. We agree with the reviewer that there may be not necessary to take so many climate extreme indices into consideration when a study does not focus on hydrological extremes. However, it is difficult to determine a one-to-one linkage between climate and hydrological variables due to the non-linearity of hydrological responses. Only a few climate variables may be not enough to describe climate conditions which have great influence on hydrological processes. The main objective of this study is to investigate the transferability of climate simulation uncertainty to hydrological impacts. The results show that including climate extreme indices improves the transferability of climate simulation uncertainty, compared with the results of Chen et al. (2016). Therefore, climate conditions described by extreme indices are found to be important for the transferability of climate uncertainty.

These issues will be discussed in the revised manuscript.

> *2.2.1 Climate Simulations; L9: Why did the authors select 50 models? The authors mentioned before that the CMIP5 archive includes 61 models. Is there a reason why the other 11 models are excluded from the selection approaches? In addition, each climate model has one or more ensemble member. Did the authors select the first ensemble member or did they select random ensemble members? The authors need to include this information in the method description, for instance by adding extra information to Table 1.*

Thanks for your comments. We actually employed 26 GCMs from the CMIP5 ensemble (Table 1) with simulations based on two Representative Concentration Pathways (RCP4.5 and RCP 8.5), with the exception of CMCC-CESM, which only used RCP8.5, and MRI-ESM1, which only used RCP4.5. On the whole, 50 climate simulations were used. Some GCMs in the CMIP5 ensemble were excluded due to lack of relevant variables (e.g., daily outputs that are necessary to drive the hydrological model) or lack of temporal coverage (e.g., the reference or future periods used in this study). In addition, GCMs employed in this study are consistent with Chen et al. (2016) to make the two studies more comparable.

In addition, only the first ensemble member of each GCM was used; this information will be clarified in the Data section.

*2.2.2 Observations; L17: Where are the data from the 100 rain gauges, 8 temperature gauges, and 1 streamflow gauge obtained? The authors have to include some references to the sources where they obtained the meteorological and discharge data.*

Thanks for the suggestion to include references to the sources of observational data. Observations in the Xiangjiang watershed are the same as those used by Zeng et al. (2016) and Xu et al. (2013) and were provided by the Changjiang Water Resources Commission. This information will be clarified in the Data section.

*3.2 Generation of Climate Scenarios: Why did the authors use the DS method to downscale GCMs and not a method such as the Advanced Delta Change approach or the Quantile Mapping approach that also take changes in extremes into consideration? It might be interesting to discuss potential uncertainties that are introduced by the downscaling approaches in the Discussion.*

Thanks for the comment. We failed to describe clear enough that Daily Scaling (DS) method is an advanced delta change approach combining delta change and quantile mapping methods. The rationale behind using a change factor method instead of a bias correction method is that climate change signals may be modified by some forms of bias correction (e.g., see Cannon et al., 2015). Still, in order to investigate the influence of alternative bias correction/downscaling approaches, we have also used the Quantile Mapping (QM) approach to post-process the CMIP5 GCMs. Figure R1 shows the results of average PSC in this case. For the Xiangjiang watershed, temperature variables were excluded in the process of selection, while they were included for the Manicouagan 5 watershed. Compared with the results calculated by the DS method, the overall characters of the results are mostly the same. The choice of the downscaling method has little influence on the conclusions of this study. However, to highlight the potential sensitivity of results to different downscaling methods, this information will be added to the Discussion section of the revised manuscript.

[Figure]

**Figure R1: The average PSC for three different selection methods over two watersheds when using the QM approach (CLI = climate variables and HYD = hydrological variables).**

*3.3.1 Hydrological Modelling; L15-21: I would recommend replacing and to discuss this part in more detail in the Results Section, for instance under a separate subsection "Calibration and Validation"*

We agree with the reviewer. The calibration and validation of the hydrological model will be presented in the Results section as follows.

The basin-averaged daily minimum and maximum temperature and precipitation in calibration and validation periods, as shown in Table R1, were used to calibrate and validate the GR4J-6 model over the two watersheds. Model parameters were obtained by the shuffled complex evolution optimization (Duan et al., 1992) based on the objective to maximize Nash-Sutcliffe Efficiency (NSE) (Nash and Sutcliffe, 1970). The optimally chosen sets of parameters yield a NSE between 0.87 and 0.93 over both watersheds. The calibrated GR4J-6 also yields small relative errors of water balance with values between -0.3% and 5.4% for calibration and validation, which demonstrate the applicability of the calibrated model over both watersheds (Table R1). Figure R2 presents the mean daily hydrographs (average discharge of each calendar day across the years) for the calibration and validation periods over two watersheds. The calibrated GR4J-6 has good performance in most of the year, with the exception that discharge is overestimated in the winter for the Manicouagan 5 watershed. In addition, since snow accumulation and snowmelt processes are not important in the Xiangjiang watershed, the GR4J model (excluding snow module) was also calibrated in this watershed. Results showed that there was little difference between the calibrated GR4J and GR4J-6 models, and thus the presence of the CemaNeige snow module would not influence the performance of GR4J-6 in the rainfall-characterized Xiangjiang watershed (Table R1).

[Figure]

**Figure R2: Observed and simulated mean hydrographs for (a, c) calibration and (b, d) validation periods over the (a, b) Xiangjiang and (c, d) Manicouagan 5 watersheds.**

**Table R1: Nash-Sutcliffe Efficiencies (NSE) and relative errors of water balance (RE) of hydrological models in the calibration and validation over two watersheds**

| Country | Watershed name | Area (km²) | Hydrological Model | Calibration period | NSE calibration | RE calibration | Validation period | NSE validation | RE validation |
|---|---|---|---|---|---|---|---|---|---|
| China | Xiangjiang | 52150 | GR4J-6 | 1975-1987 | 0.912 | -0.3% | 1988-2000 | 0.871 | 5.4% |
| | | | GR4J | 1975-1987 | 0.912 | -0.2% | 1988-2000 | 0.872 | 5.5% |
| Canada | Manicouagan 5 | 24610 | GR4J-6 | 1970-1979 | 0.926 | 3.8% | 1980-1989 | 0.881 | 2.7% |

*3.4 Data Analysis: It would be good to an additional sentence on what it means when a PSC is high or low. For the reader, it might be more difficult to image the meaning of a high or low PSC.*

Thanks for your suggestion. A higher PSC means that the selected subset covers a larger uncertainty range. This will be clarified in the revised manuscript.

*Figure 1: The longitude axes are given, but the latitude axes are missing. Further I think the figure does not contain a lot of information. I would add a digital elevation model or another topographic/geographic info to give the reader more valuable information on the characteristics of the catchments. In addition, I would add the positions of the discharge gauging stations used for the calibration/validation. Finally, I recommend making inlets,*

*including the catchment maps, larger.*

We agree with the reviewer. Figure R3 will be updated in the revised manuscript.

[Figure]

**Figure R3: Location maps of the (a) Xiangjiang and (b) Manicouagan 5 watersheds (The study area in the Xiangjiang watershed is one of its sub-basins as the orange boundary shows).**

*Technical Comments*

*Abstract; L16: ". . .the importance of choosing climate variables properly while selecting subsets. . ." instead of ". . .the importance of properly choosing climate variables in selecting subsets. . ."*

This sentence will be revised as suggested.

**References**

Cannon, A. J., Sobie, S. R., and Murdock, T. Q.: Bias Correction of GCM Precipitation by Quantile Mapping: How Well Do Methods Preserve Changes in Quantiles and Extremes?, Journal of Climate, 28, 6938-6959, https://doi.org/10.1175/JCLI-D-14-00754.1, 2015.

Chen, J., Brissette, F. P., and Lucas-Picher, P.: Transferability of optimally-selected climate models in the quantification of climate change impacts on hydrology, Climate Dynamics, 47, 3359-3372, https://doi.org/10.1007/s00382-016-3030-x, 2016.

Duan, Q., Sorooshian, S., and Gupta, V.: Effective and efficient global optimization for conceptual rainfall-runoff models, Water Resources Research, 28, 1015-1031, https://doi.org/10.1029/91WR02985, 1992.

Knutti, R., Furrer, R., Tebaldi, C., Cermak, J., and Meehl, G. A.: Challenges in Combining Projections from Multiple Climate Models, Journal of Climate, 23, 2739-2758, https://doi.org/10.1175/2009jcli3361.1, 2010.

Mendlik, T., and Gobiet, A.: Selecting climate simulations for impact studies based on multivariate patterns of climate change, Climatic Change, 135, 381-393, https://doi.org/10.1007/s10584-015-1582-0, 2016.

Nash, J. E., and Sutcliffe, J. V.: River flow forecasting through conceptual models part I — A discussion of principles, Journal of Hydrology, 10, 282-290, https://doi.org/10.1016/0022-1694(70)90255-6, 1970.

Reifen, C., and Toumi, R.: Climate projections: Past performance no guarantee of future skill?, Geophysical Research Letters, 36, https://doi.org/10.1029/2009gl038082, 2009.

Xu, H., Xu, C.-Y., Chen, H., Zhang, Z., and Li, L.: Assessing the influence of rain gauge density and distribution on hydrological model performance in a humid region of China, Journal of Hydrology, 505, 1-12, https://doi.org/10.1016/j.jhydrol.2013.09.004, 2013.

Zeng, Q., Chen, H., Xu, C.-Y., Jie, M.-X., and Hou, Y.-K.: Feasibility and uncertainty of using conceptual rainfall-runoff models in design flood estimation, Hydrology Research, 47, 701-717, https://doi.org/10.2166/nh.2015.069, 2016.

---

## Author Comment (AC2) · 29 Mar 2018

Dear Anonymous Referee #2,

We would like to thank the reviewer for the time taken in reviewing this paper. All comments will be incorporated into the revised manuscript. Please check the attached point-by-point replies. We will make the revisions to the manuscript as suggested.

Please also note the supplement to this comment: https://www.hydrol-earth-syst-sci-discuss.net/hess-2017-703/hess-2017-703-AC2-supplement.pdf
* * *
703, 2018.

**Supplement:**

Replies to Referee #2

**Transferability of climate simulation uncertainty to hydrological climate change impacts**

Hui-Min Wang, Jie Chen, Alex J. Cannon, Chong-Yu Xu, Hua Chen

We would like to thank the reviewer for the time taken in reviewing this paper. All comments will be incorporated into the revised manuscript. Please find the point-by-point responses below. For clarity, comments are given in italics, and our responses are given in plain text. We will make the revisions to the manuscript as suggested.

> *The authors do a good job in their attempt to shed light on the important problem that impact modelers face in efficiently and effectively capturing the range of uncertainty in climate model simulations. Furthermore, they investigate whether covering this range in climate model output variables translates to capturing the uncertainty range of hydrological variables. The paper is well written and clearly presented. Though, in the end, I was not convinced that impact modelers can actually save much time and effort using this methodology. I would recommend that the manuscript needs minor revisions. Importantly, the authors need to make it clearer how an end user can avoid downloading all 50 simulations in order to prove which subset of 10 are most appropriate to cover the uncertainty range in their study.*

Thanks for your positive evaluation in general and for your professional comments. Please find our responses in next page.

> *I would begin by asking this. What do end users or impact modelers gain by this paper? You have shown that for your two different watersheds, a subset of approximately 10 model simulations are needed to reasonably capture the spread of the model uncertainty for both climate and hydrological variables. Additionally, you point out that not using the temperature variables to obtain the subset affects the hydrology of the two watersheds differently. As a result, you illustrate that the selection of the 10 climate models is unique to each impact assessment study. Furthermore, you needed all 50 simulations to test which subset was optimal for your two cases. I do not see how an impact modeler would not need to repeat precisely what you have done. In order to replicate your method, but specific to their study interest or area, they would need to "extract, store, and compute" (page 2, line 14) all 50 model simulations themselves. Then, couldn't they just as easily implement the entire set of simulations instead of a smaller subset? To ask it more directly: How can an impact modeler know which 10 model simulations to use, for their unique case, without testing the ensemble ranges of each possible subset with respect to the entire set of simulations? And to do this, would they not need to run*

*all 50 model simulations?*

Sorry for the lack of clarity in the manuscript. Depending on the choice of climate variables and climate model ensemble, it may not be necessary to "extract, store, and compute" climate indices from all climate model simulations in the ensemble of interest. For example, pre-computed ETCCDI climate extreme indices for GCMs participating in CMIP3 and CMIP5 are publically available from http://climate-modelling.canada.ca/climatemodeldata/climdex/. In addition, the end-user may be able to refer the results of this study for watersheds with similar climate and hydrological characteristics, although it is likely that some level of site and study-specific analysis will be required.

However, the main objective of this study is to investigate the transferability of climate simulation uncertainty to hydrological impacts. If the climate simulation uncertainty is transferable in the hydrological impacts, the selected 10 climate simulations can be directly used to drive a hydrological model for impacts studies instead of using all climate simulations. This is crucial for hydrological modelers as they usually spend a lot of computational costs in running a large number of climate simulations with a complicated hydrological model (e.g. SWAT). The conclusion of this study shows that the climate simulation uncertainty is transferable in the envelope-based selection based on multiple climate variables, and the subset of around 10 climate simulations can cover the majority of uncertainty. Therefore, end-users can choose the group of climate variables according to their knowledge to the climate and hydrological characteristics of watershed of interest and then select the representative subset of climate simulations to save computational costs in the hydrology world.

All above information will be discussed in the Discussion section of the revised manuscript.

*Some more specific comments and questions are as follows:*

*In section "2.2.1 Climate Simulations": Does it make sense to lump the uncertainty ranges of both RCP4.5 and RCP8.5? These are two different concentration pathways that represent very different conditions. It is true that we currently can't know which is more likely. I would recommend either treating each pathway independently with different ranges of uncertainty, or I would recommend also including simulations from pathways RCP2.6 and RCP6.*

Thanks for your comment. RCP4.5 is the medium stabilization scenario and RCP8.5 represents the very high radiative forcing scenario. The mitigation scenario, RCP2.6, was not used because recent analyses suggest that this RCP will be very difficult to achieve with current emission trajectories (Arora et al., 2011; Rozenberg et al., 2015). RCP6.0 is a scenario with radiative forcing that is bracketed by RCP4.5 and RCP8.5 and was not simulated by as many modeling centers as RCP4.5 and RCP8.5. Thus, we used RCP4.5 and RCP8.5 to include a range of realistic projections (Lutz et al., 2016). Due to unknown future emission scenarios, two concentration pathways were used in an

undifferentiated manner to cover uncertainty resulting from emission scenarios in our study.

We agree with the reviewer that two emission scenarios generate respective climate simulations. It may be more proper to separately use RCPs in the practical applications. Thus, we have also used each pathway separately to be input into the subset selection. Figure R1 shows the example results where only the one scenario (RCP4.5 or RCP8.5) was used in the Xiangjiang watershed (temperature variables were not included in the selection process). The main characters of the results are roughly the same as the original research where 2 RCPs were considered. In this case, the selection of 5 or 6 climate simulations by the KKZ method can cover adequate uncertainty range. Therefore, the specific choice of emission scenarios can be decided by end-users according to their own needs. The results for the Manicouagan 5 watershed will be further explored in the future.

In order to stress this comment, an explanation on the choice of emission scenarios will be added in the Discussion section.

[Figure]

**Figure R1: The average PSC for three different selection methods over the Xiangjiang watersheds when only one emission scenario is considered (CLI = climate variables and HYD = hydrological variables)**

*Page 7, line 17: What was the reason to use 100 quantiles instead of the total number of days in the periods (e.g., 1975-2004 for January = 30 years times 31 days = 930 days or quantiles)?*

The use of 100 quantiles is to smooth the distribution of simulated daily precipitation or temperature. The smoothing process eliminates sharp scaling factors that may occur due to outliers, especially for extreme values. On the other hand, Lafon et al. (2013) found that the division of 100 quantiles in the empirical quantile mapping generates more accurate downscaling results than that of 25, 50 or 75 quantiles. The use of 100 quantiles in the Daily Scaling (DS) method is also the same as many other studies (Harrold and Jones, 2003; Mpelasoka and Chiew, 2009; Chen et al., 2013). All of these points will be clarified in the revised manuscript.

*Page 9, line 5: I do not anticipate for it to change your results that much, but perhaps it is better to use something like standard deviation as a measure of the uncertainty coverage. The*

*Percentage of Spread Coverage (PSC) is only sensitive to the range of the minimum and maximum values. You could end up having many of the models grouped close together, and as a result, your measure would overestimate your actual uncertainty coverage.*

We agree with the reviewer that the spread of a MME provides an imperfect estimate of uncertainty, and the PSC is sensitivity to the maximum and minimum values. We considered and found that it is improper to use standard deviation, in our case, as the measure of uncertainty in the evaluation on the uncertainty coverage of subsets. To be specific, the selected simulations in impact studies are often considered to be representative of specific uncertainty range instead of individual samples in the calculation of deviation. For example, selected simulations are regarded as the 10th and 90th quantiles in the range of temperature or precipitation in many impact studies (Lutz et al., 2016; Immerzeel et al., 2013; Sorg et al., 2014).

In consideration of reviewer's concern, i.e. to lower the influence from the maximum and minimum values, the average coverage on quantiles (ACQ) of the subsets have been used as an evaluation criterion. The ACQ is calculated using Eq.(R1)

$$ACQ = \frac{1}{N}\sum_{i=1}^{N}\left(\max_s Q_{s,i} - \min_s Q_{s,i}\right) \tag{R1}$$

where $N$ is the total number of climate or hydrological variables, and $Q_{s,i}$ represents the rank of quantile of the $i$th variable for the $s$th selected simulation (e.g. if the change in a variable of one selected simulation is the 80th quantile of the changes of all simulations, then $Q_{s,i} = 0.8$). ACQ evaluates the range of quantiles covered by selected simulations. Due to the use of quantiles instead of values, the ACQ is less influenced by the maximum or minimum values. Figure R2 presents the ACQ for climate variables and hydrological variables (temperature variables were excluded in the selection for the Xiangjiang watershed, while they were included for the Manicouagan 5 watershed). Compared with average PSC, ACQ results show similar characteristics but less sensitivity, and PSC

[Figure]

**Figure R2: The average coverage on quantiles (ACQ) for three different selection methods over two watersheds (CLI = climate variables and HYD = hydrological variables)**

can more intuitively provide an evaluation of the ability of subsets to cover uncertainty. Therefore, PSC will be still used as the evaluation criterion in the revised manuscript.

This will be discussed in the revised manuscript.

> *Figure 2: Are you showing the observed and simulated values for the calibration and validation for 1 year? Or is each day the average of that day across the years (e.g., for Xiangjiang: all January 1 values are averaged over the time period 1975-1987, then January 2 values are averaged over the same years, . . .)?*

The mean hydrographs showed in Fig.2 were calculated as the average of each calendar day across the years. This will be clarified in the revised manuscript.

**References**

Arora, V. K., Scinocca, J. F., Boer, G. J., Christian, J. R., Denman, K. L., Flato, G. M., Kharin, V. V., Lee, W. G., and Merryfield, W. J.: Carbon emission limits required to satisfy future representative concentration pathways of greenhouse gases, Geophysical Research Letters, 38, https://doi.org/10.1029/2010GL046270, 2011.

Chen, J., Brissette, F. P., Chaumont, D., and Braun, M.: Performance and uncertainty evaluation of empirical downscaling methods in quantifying the climate change impacts on hydrology over two North American river basins, Journal of Hydrology, 479, 200-214, https://doi.org/10.1016/j.jhydrol.2012.11.062, 2013.

Harrold, T. I., and Jones, R. N.: Generation of rainfall scenarios using daily patterns of change from GCMs, in: Water Resources Systems - Water Availability and Global Change, edited by: Franks, S., Blöschl, G., Kumagai, M., Musiake, K., and Rosbjerg, D., 280, IAHS Press, 165-172, 2003.

Immerzeel, W. W., Pellicciotti, F., and Bierkens, M. F. P.: Rising river flows throughout the twenty-first century in two Himalayan glacierized watersheds, Nature Geoscience, 6, 742-745, https://doi.org/10.1038/ngeo1896, 2013.

Lafon, T., Dadson, S., Buys, G., and Prudhomme, C.: Bias correction of daily precipitation simulated by a regional climate model: a comparison of methods, International Journal of Climatology, 33, 1367-1381, https://doi.org/10.1002/joc.3518, 2013.

Lutz, A. F., Immerzeel, W. W., Kraaijenbrink, P. D., Shrestha, A. B., and Bierkens, M. F.: Climate Change Impacts on the Upper Indus Hydrology: Sources, Shifts and Extremes, PLoS One, 11, e0165630, https://doi.org/10.1371/journal.pone.0165630, 2016.

Mpelasoka, F. S., and Chiew, F. H. S.: Influence of Rainfall Scenario Construction Methods on Runoff Projections, Journal of Hydrometeorology, 10, 1168-1183, https://doi.org/10.1175/2009jhm1045.1, 2009.

Rozenberg, J., Davis, S. J., Narloch, U., and Hallegatte, S.: Climate constraints on the carbon intensity of economic growth, Environmental Research Letters, 10, 95006, https://doi.org/10.1088/1748-9326/10/9/095006, 2015.

Sorg, A., Huss, M., Rohrer, M., and Stoffel, M.: The days of plenty might soon be over in glacierized Central Asian catchments, Environmental Research Letters, 9, 104018, https://doi.org/10.1088/1748-9326/9/10/104018, 2014.

---

## Author Response (AR2)

Authors' responses to comments

**Transferability of climate simulation uncertainty to hydrological impacts**

Hui-Min Wang, Jie Chen, Alex J. Cannon, Chong-Yu Xu, Hua Chen

We sincerely appreciate the editor's valuable suggestions which helped to improve the manuscript. We have responded to all comments point-by-point as follows, and the manuscript has been revised accordingly. For clarity, all comments are given in italics and responses are given in plain text.

**Responses to Editor's comments**

*From R1 response: Please limit the information on the background maps of Figure 1 to the minimum needed for geographic orientation: i.e. continental land masses, no political borders.*

Thank you for the suggestion on the presentation of Fig.1. The background maps in Figure R1 have been modified and updated in the revised manuscript [Fig.1].

[Figure]

**Figure R1: Location maps of the (a) Xiangjiang and (b) Manicouagan 5 watersheds (The study area in the Xiangjiang watershed is one of its sub-basins as the orange boundary shows).**

*From R2 response: Some of the response to the raised concerns of the aims and the question of saving future analysis, which you answered in detail in the response should also find their place in the manuscript. My suggestion is to sharpen and clarify the aims/objectives of 'transferable uncertainty' more prominently in the introduction and to discuss the question of 'savings vs redo' in the discussion section.*

We appreciate the editor's suggestion to address some of R2 responses in the manuscript. Some modifications have been applied in the Introduction section to sharpen the aim of this study [Lines 18-20 and Lines 24-26, Page 3]. In addition, the benefits that impact modellers can gain from this

study were added in the Discussion [Lines 21-29, Page 14], and more study-specific analysis in the future was also suggested [Lines 12-17, Page 15].

[revised manuscript text omitted]